# Genome-Wide Identification and Expression Analysis of the PLATZ Transcription Factor in Tomato

**DOI:** 10.3390/plants12142632

**Published:** 2023-07-13

**Authors:** Lifang Zhang, Tao Yang, Zepeng Wang, Fulin Zhang, Ning Li, Weijie Jiang

**Affiliations:** 1College of Horticulture, Xinjiang Agricultural University, Urumqi 830052, China; zlf1225@foxmail.com (L.Z.); zx1119802932@163.com (Z.W.); 18095998056@163.com (F.Z.); 2Key Laboratory of Genome Research and Genetic Improvement of Xinjiang Characteristic Fruits and Vegetables, Institute of Horticultural Crops, Xinjiang Academy of Agricultural Sciences, Urumqi 830091, China; yangtao_xj@sina.com; 3Institute of Vegetables and Flowers, Chinese Academy of Agricultural Sciences, Beijing 100081, China

**Keywords:** *PLATZ* gene family, phylogeny analysis, expression analysis, biotic/abiotic stresses

## Abstract

The PLATZ (plant AT protein and zinc-binding protein) transcription factor family is involved in the regulation of plant growth and development and plant stress response. In this study, 24 *SlPLATZs* were identified from the cultivated tomato genome and classified into four groups based on the similarity of conserved patterns among members of the same subfamily. Fragment duplication was an important way to expand the *SlPLATZ* gene family in tomatoes, and the sequential order of tomato *PLATZ* genes in the evolution of monocotyledonous and dicotyledonous plants and the roles they played were hypothesized. Expression profiles based on quantitative real-time reverse transcription PCR showed that *SlPLATZ* was involved in the growth of different tissues in tomatoes. *SlPLATZ21* acts mainly in the leaves. *SlPLATZ9*, *SlPLATZ21*, and *SlPLATZ23* were primarily involved in the red ripening, expanding, and mature green periods of fruit, respectively. In addition, *SlPLATZ1* was found to play an important role in salt stress. This study will lay the foundation for the analysis of the biological functions of *SlPLATZ* genes and will also provide a theoretical basis for the selection and breeding of new tomato varieties and germplasm innovation.

## 1. Introduction

An important class of transcriptional regulatory molecules known as transcription factors (TFs) play a role in the control of numerous biological processes, including development, signaling, and stress response. For instance, NF-Y, MYB, and WRKY TFs play significant roles in various stress tolerances, while NAC and GRF govern the creation of roots, flowers, and seeds. SPL TFs are also engaged in the transformation of young plants into adult plants [1]. Gene transcription can be triggered or suppressed by TF binding to *cis*-acting regions of target genes. The expression of numerous genes can be regulated by transcription factors, which typically have only one or more regions that bind to DNA. However, a small number of transcription factors have been shown to control the expression of a single gene [2]. Therefore, a thorough study of transcription factors will enable us to comprehend their evolutionary background, biological activities, and regulatory mechanisms [3].

The plant AT-rich sequence and zinc-binding (PLATZ) TF family is a novel class of plant-based zinc ion and DNA-binding proteins [4]. By attaching to cis-acting components in the upstream promoter region, DNA-binding proteins participate in a variety of biological processes, including DNA replication, the control of gene expression, etc. [5]. It has been established that the pea protein PLATZ1 binds to the upstream DE1 element at the A/T-enriched DNA sequence, which is required for transcriptional repression [4]. Several plant species have been shown to include members of the PLATZ family, including 13 genes in the Arabidopsis genome, 8 genes in Glycine max (soybean), 9 genes in Gossypium hirsutum (cotton), 15 genes in Oryza sativa (rice), and 17 genes in Zea mays (maize). PLATZ proteins play an important role in plant growth and development and abiotic stress responses.

PLATZ proteins play a role in plant growth, development, and senescence. In cotton, PLATZ1 mediates the signaling pathways for ABA, GA, and ethylene and is involved in seed germination and seedling establishment [6]. The PLATZ family of transcription factors is also crucial for the growth and development of leaves. ORESARA15 is a transcription factor of the PLATZ TF family, and this gene not only enhances leaf growth by promoting the rate and duration of cell proliferation at an early stage but also regulates the GROWTH-REGULATING FACTOR (GRF)/GRF-INTERACTING FACTOR signaling pathway, thereby inhibiting leaf senescence at a later stage [7]. The formation and growth of seeds are also significantly influenced by the PLATZ transcription factor family. In rice, GL6 is a plant-specific PLATZ TF that interacts with C53 to influence rice grain development and negatively control grain number [8]. Instead of being expressed in the endosperm, the *SG6* gene encodes a PLATZ protein that is extensively distributed in cells and is primarily expressed in the early stages of embryonic development. The division of rice spikelet hull cells, which determine grain size, is controlled by the rice (*Oryza sativa*) mutant shortgrain6 (*sg6*). Overexpression of *SG6* results in larger and heavier grains, as well as increased plant height [9]. Additionally, *AtPLATZ7* is essential for RGF1 signaling, which controls the growth of root meri-stem tissue through ROS signaling [10]. The maize genome contains 15 *PLATZ* genes (*ZmPLATZ*), which are crucial for RNAP III-mediated transcriptional control [11,12]. RPC53 and TFC1 of the RNAP III transcriptional complex interact with FL3 (*ZmPLATZ12*) to control the expression of tRNA and 5 S rRNA, which control the growth and development of the kernel in maize seeds [11]. When developing poplar stems, a PLATZ protein is crucial for the change from primary to secondary growth [13]. In a comparative transcriptome study of normal lint (FL) and congenic non-lint (Fl) diploid cotton fibers, the results showed that a PLATZ transcription factor of Fl was down-regulated after 10 d of flowering compared to FL, indicating that PLATZ is crucial for the formation and extension of cotton fibers [14]. There is speculation that the *PLATZ* gene, which has variable expression in mature and immature sugarcane tissues with high fiber genotypes, may be involved in the transcriptional regulation of secondary cell wall formation [15]. Mutants of *ZmPLATZ12* caused the endosperm to appear flocculent, indicating that *ZmPLATZ12* plays a significant role in the development of the endosperm and the filling of storage material [11]. In the endosperm, *ZmPLATZ2* is highly expressed and can bind to the promoters of *ZmSSI*, *ZmISA1*, and *ZmISA2* to boost the expression of those genes. Four starch synthesis-related genes, including *OsSSI*, were significantly upregulated in transgenic plants by *ZmPLATZ2* overexpression in rice. The *PLATZ* gene GRMZM2G31165 (*ZmPLATZ2*) can bind to the promoter region of key genes for starch synthesis under glucose induction and can promote starch synthesis by positively regulating the Glu pathway [16].

PLATZ transcription factors have a role in the adversity-stress response in addition to being crucial for plant growth and development. In Arabidopsis, *AtPLATZ1* and *AtPLATZ2* were reported to be positive regulators of seed desiccation tolerance. Asexual tissues become partially seed dehydration-tolerant when *AtPLATZ1* is overexpressed in wild-type Arabidopsis, which enhances their capacity to consume water and endure drought [17]. *AtPLATZ2* is a transcription factor that negatively regulates salt tolerance in Arabidopsis thaliana by repressing the expression of genes such as *CBL4/SOS3* and *CBL10/SCaBP8* [18]. Due to the transfer of the cotton gene *GhPLATZ1*, the seed germination rate and seedling survival of GhPLATZ1 transgenic Arabidopsis thaliana were significantly higher under salt and mannitol stress compared with the wild type, demonstrating increased resistance to osmotic stress [6]. *GmPLATZ1* is involved in the germination of soybean under osmotic stress [19]. In maize, the expression of PLATZ1 homologs was significantly up-regulated by drought stress, indicating that PLATZ genes play an important role in drought resistance in maize [20].

The tomato (*Solanum lycopersicum* L.) is one of the most widely cultivated and economically valuable vegetable crops in the world and has great potential for application. Long-term exposure of tomatoes to adversities such as drought, high salt, low temperature, and high temperature has a significant impact on their quality and productivity. Salinity is one of the main causes of crop yield reduction, which also affects plant physiology and results in vinylation, wilting, and biochemical alterations [21]. Therefore, molecular and cellular, physiological, and biochemical perspectives are needed to cope with and adapt to environmental stresses [22]. The availability of the whole tomato genome sequence, along with the genome sequences of other species including Arabidopsis, potato, and tobacco, has made it possible to uncover tomato PLATZ family genes thanks to the advancement of tomato genomics. Consequently, the identification of superior tomato genes should advance tomato research and, as a result, help to better direct tomato production.

In this study, we performed a systematic and comprehensive genome-wide analysis of the PLATZ TF protein of tomatoes. After extensive research on tomato *PLATZ* gene family members, 24 and 13 *PLATZ* genes, respectively, were found in cultivated tomatoes and wild tomato pennellii. 24 SlPLATZs and 13 spPLATZs were thoroughly studied in terms of chromosomal localization, gene structure, conserved patterns, phylogenetic analysis, and tandem and selective pressures. The evolutionary relationships of tomato PLATZs with Arabidopsis thaliana, pepper, tobacco, rice, maize, Solanum melongena, and pennellii were also examined. Quantitative real-time polymerase chain reaction (qRT-PCR) was used to determine the different tissue expression profiles of SlPLATZs in tomatoes. Additionally, it was investigated how SlPLATZs were expressed relative to each other in the roots, stems, and leaves. And the relative expression of 3 *SlPLATZ* gene family representative genes at the expanding, mature green, breaker, and red ripening periods under the application of various amounts of NAA, EBR, and MT treatments. This study will provide a useful resource for further studies to understand the regulation of PLATZ proteins on development and stress resistance in tomatoes.

## 2. Results

### 2.1. Identification and Chromosomal Localization of PLATZ Gene Family Members in Tomato

Following genome-wide searches to identify possible sequences in pennellii and cultivated tomatoes, the structural domain integrity was manually confirmed in the database, which yielded 24 *PLATZ* genes in cultivated tomatoes and 13 *PLATZ* genes in pennellii tomatoes. All pennellii tomato *PLATZ* genes were named *SpPLATZ1* to *SpPLATZ13*, and the cultivated tomato *PLATZ* genes were named *SlPLATZ1* to *SlPLATZ24* based on their physical location and chromosome order (Appendix A). The 24 *SlPLATZ* proteins had a wide range of lengths, MWs, and pIs, with an average amino acid number of 181 encoded by the *SlPLATZ* gene sequences. These amino acid counts ranged from 90 (SlPLATZ12) to 255 (SlPLATZ18). A molecular weight range of 10.38 kDa (SlPLATZ12) to 29.16 kDa was observed (SlPLATZ18). The isoelectric point was between 6.19 (SlPLATZ19) and 9.54 (SlPLATZ17). Twenty-three of these *SlPLATZ* proteins had isoelectric points greater than 7, which indicated that these proteins were primarily basic. All SlPLATZ proteins, except SlPLATZ12, had GRAVY values less than 0, which indicated that they were hydrophilic proteins. One of the 24 *PLATZ* transcription factor families was distributed throughout the mitochondrial matrix; one member was located in the Golgi apparatus, nine members were located in the cytoplasm; and the remaining 13 members were located in the nucleus.

The 24 *SlPLATZ* genes were irregularly distributed on 9 chromosomes. Thirteen *SlPLATZ* genes were located on chromosome 2, followed by 3 genes on chromosome 7, 2 genes on chromosome 8, and 1 gene on chromosomes 1, 3, 4, 6, 10, and 12. *SlPLATZ15*, *SlPLATZ16*, *SlPLATZ18*, *SlPLATZ22*, and *SlPLATZ24* were scattered near the heads of their respective chromosomes, and *SlPLATZ1*, *SlPLATZ17*, *SlPLATZ19*, *SlPLATZ20*, *SlPLATZ21*, and *SlPLATZ23* were distributed near the ends of their respective chromosomes (Figure 1).

### 2.2. Analysis of the Structure, Conserved Domain, and Motif of the SlPLATZ Gene

The structural diversity of the family members is reflected in the variation in the number of introns and exons in genes belonging to the same family. The exon and intron architectures of the *SlPLATZ* genes were distinct in different subfamilies and conserved within the same subfamily, according to statistics on the tomato *PLATZ* gene family. The number and length of introns fluctuated significantly between various subfamilies, as depicted in the figure. The number of exons ranged from 1–4, and the number of introns ranged from 0–4. *SlPLATZ1*, which had 4 exons and 4 introns, had the most introns and exons. *SlPLATZ17*, *SlPLATZ21*, and *SlPLATZ22* are architecturally extremely similar and have three exons and three introns (Figure 2B). *SpPLATZ3* and *SpPLATZ4* on chromosome 2 have three exons and are structurally very similar; the same applies to *SpPLATZ7*, *SpPLATZ10*, and *SpPLATZ11* (Appendix A). The intron position and length varied between members.

Highly conserved DNA sequences may have the same utility as molecular sequences that are extremely similar, homogeneous, or conserved. The conserved motifs of members of the same subfamily were similar, according to an examination of the conserved motifs of tomato *PLATZ* gene family members. The MEME online program identified 10 motifs, designated motif 1 to motif 10, and analyses showed a high degree of similarity in the conserved motifs of the *SlPLATZ* gene family. Every *SlPLATZ*, except for *SlPLATZ12*, has motif 1, and the majority of *SlPLATZ*s also have motif 2. According to the investigation, the SlPLATZ family belonged to a group whose amino acid motif composition was roughly similar. We observe that the conserved motifs in *SlPLATZ17*, *SlPLATZ21*, and *SlPLATZ22* are comparable, and the number of exons and introns is likewise consistent. The same pattern is also present in *SlPLATZ18* and *SlPLATZ24* (Figure 2A). All *SpPLATZ* have motif1, motif2, and motif5, with motif5 dispersed across the protein’s C-terminus. *SpPLATZ7*, *SpPLATZ10*, and *SpPLATZ11* in Pennellii shared conserved motifs, while *SpPLATZ8* and *SpPLATZ13* also displayed this pattern (Appendix A).

### 2.3. Phylogenetic Analysis of the PLATZ Family

Varying *PLATZ* gene counts have been found in domesticated and wild Solanaceae species, including cultivated tomatoes, wild tomatoes, and potatoes (24, 13, and 18, respectively). The *PLATZ* gene sequences from Arabidopsis (12), potato (18), rice (13), maize (14), soybeans (31), cotton (41), wheat (46), sorghum (23), and the 24 *SlPLATZ* genes and 13 *SpPLATZ* genes were studied phylogenetically to further understand the genetic evolutionary relationship of PLATZ. Based on the position of Arabidopsis in the phylogenetic tree, *PLATZ* was divided into six groups, named I-VI. Groups I, II, III, IV, and VI contained monocotyledons and dicotyledons, which indicated that the PLATZ members of these subfamilies were more conserved family members. Group V exclusively contains monocotyledons, and Group VI only contains dicotyledons, which indicated that PLATZ family members expanded with the differentiation of monocotyledons and dicotyledons but are related to dicotyledons in the completion of their complex life activities. Different branches existed within each subgroup, and potatoes, pennellii tomatoes, and cultivated tomatoes were grouped in Groups I, II, IV, and VI but not with Arabidopsis, which indicated divergence of the Solanaceae. Group I contained five *SpPLATZ* and fifteen *SlPLATZ* genes; Group II contained four *SpPLATZ* and four *SlPLATZ* genes; Group IV contained one *SpPLATZ* and one *SlPLATZ* gene; and Group VI contained three *SpPLATZ* and three *SlPLATZ* genes. Both cultivated and wild tomatoes have extremely homologous *PLATZ* genes in all subfamilies. The fact that there were three times as many cultivated tomato *SlPLATZ* genes in Group I as the pennellii tomato *SpPLATZ* genes suggests that cultivated tomatoes descended from pennellii tomatoes (Figure 3).

### 2.4. Analysis of Collinearity and Selection Pressure on the Tomato PLATZ Gene Family

MCScanX was used to analyze the synteny of the *SlPLATZ* genome. The *SlPLATZ* gene family in tomatoes appears to have grown via fragment duplication, as evidenced by the presence of the paralogous gene pairs *SlPLATZ21* and *SlPLATZ22*, *SlPLATZ18* and *SlPLATZ24*, and *SlPLATZ17* and *SlPLATZ21* (Figure 4). There are four gene pairs found in wild tomato pennellii: *SpPLATZ7* and *SpPLATZ10*, *SpPLATZ7* and *SpPLATZ11*, *SpPLATZ8* and *SpPLATZ13*, and *SpPLATZ10* and *SpPLATZ11*. Except for the tandem duplication of *SpPLATZ10* and *SpPLATZ11*, the remaining gene pairs are segmental (Appendix A). All collinear gene pairings had nonsynonymous (Ka) to synonymous (Ks) substitution ratios (Ka/Ks) that were less than 1, which showed that purifying selection mostly affected *SlPLATZ* genes, and these genes had slow evolutionary rates and stable gene functions. The divergence times of the three gene pairs were inferred based on Solanaceae’s differentiation rate R (1.5 × 10^−8^), and the results revealed that the newest (*SlPLATZ17/SlPLATZ21*) and oldest (*SlPLATZ21/SlPLATZ22*) pairs diverged at 26.600 and 31.767 Mya, respectively (Appendix A). The replication times of *SpPLATZ* homologous gene pairs in pennellii ranged from 20.813 to 29.376 Mya (Appendix A). These divergence times also matched the findings of their evolutionary rates.

An analysis of the collinearity diagram revealed that seven *SlPLATZ* genes on tomato chromosomes 1, 2, 4, 6, and 8 were collinear with seven corresponding genes in Arabidopsis (Figure 5). Only two rice homolog genes and two tomato-specific genes, *SlPLATZ2* on chromosome 2 and *SlPLATZ18* on chromosome 7, were co-linear, which suggests that *SlPLATZ2* and *SlPLATZ18* are conserved *PLATZ* genes that are shared by monocots and dicots and descended from a common ancestor. We also discovered that *SlPLATZ2* and *SlPLATZ18* were distributed in Group II but not on the same branch in the phylogenetic tree, which indicated that the genes in Group II served the original purpose (Figure 3). However, the gene groups in Group VI had new functions that developed from *PLATZ* genes throughout evolution. *SlPLATZ17* on chromosome 6, *SlPLATZ21,* and *SlPLATZ22* on chromosome 8, and their homologs in Arabidopsis thaliana, were mutually multicollinear genes with similar functions. This result is consistent with the collinearity between *SlPLATZ* species. *SlPLATZ17*, *SlPLATZ21*, and *SlPLATZ22*, which are directly homologous in tomatoes, are likely genes that develop during dicotyledonous plant differentiation and play a significant role in *PLATZ* gene evolution. *SlPLATZ17*, *SlPLATZ21*, and *SlPLATZ22* from cultivated tomatoes and *AT4G17900*, *AT5G46710*, and *AT1G32700* from Arabidopsis are located on the same branch in Group IV of the phylogenetic tree. Therefore, it is hypothesized that *AT4G17900*, *AT5G46710*, and *AT1G32700* are orthologous genes in Arabidopsis with similar functions.

Many monocotyledonous and dicotyledonous *PLATZ* genes are found in Group I (Figure 3). *SlPLATZ17*, *SlPLATZ21*, and *SlPLATZ22* underwent quantitative amplification rather than functional differentiation to support the complex life activities of dicotyledonous plants. 8 *SlPLATZ* genes and 6 tobacco homologs were collinear, and 10 *SlPLATZ* genes and 10 potato homologs were collinear. *SlPLATZ2* and 1 potato homolog were collinear, and *SlPLATZ18* and 2 potato homologs were collinear. These results indicated that *SlPLATZ2* evolved conservatively during Solanaceae history and that *SlPLATZ18* underwent expansion, which further demonstrated that *SlPLATZ2* and *SlPLATZ18* were conserved genes. The physical locations of the collinear genes for cultivated tomato and potato on their respective chromosomes were identical, but the physical locations of the collinear genes for cultivated tomato and tobacco were different. Collinearity was similar between pennellii and other species, except for variations in rice and maize, according to the collinearity diagram of cultivated tomatoes and other species (Appendix A). This result means that the whole tomato gene family in Solanaceae did not constrict or expand from its current condition, and the evolution of the *PLATZ* family genes was unaffected by the separation of the cultivated tomato from the potato and the cultivated tomato from pennellii.

### 2.5. Analysis of Cis-Acting Regulatory Elements of SlPLATZ Genes

*Cis*-regulated elements (CREs) are segments of DNA sequences in gene promoters that affect gene expression. The promoter sequences of 24 *SlPLATZ* family members contain cis-acting elements that are associated with various types of stress responses, hormone responses, development, cell cycle, and transcription, including CCAAT-boxes, TC-rich repeats, AREs, G-boxes, Box 4, and other elements associated with stress responses (Figure 6). GCN4_motifs, CAT-boxes, and other growth and developmental response elements were found. ABREs, CGTCA-motifs, TGACG-motifs, TGA-elements, and other hormone response elements were found. Circadian, MSA-like, and other cell cycle response elements were identified. A-box, AT-rich element, and other transcriptional response elements were identified. The light response element Box 4 and the anaerobic induction element ARE were present in 24 *SlPLATZ* genes under adverse stress. The two stress-related cis-acting elements, G-Box and Box 4, were distributed in all upstream sequences of *SlPLATZ* genes, with up to 109 and 171, respectively, and Box 4 in *SlPLATZ17* was the most abundant element, with 14. *SlPLATZ14* and *SlPLATZ15* had the most G-Boxes, with 9. With 25 occurrences in 24 *SlPLATZ* genes and accounting for 45% of the growth response *c*is-acting elements, seed-specific control of the O2-site was the most prevalent *c*is-acting element in growth development. Abscisic acid-responsive elements (ABREs), which made up 30% of the hormone response *cis*-acting elements and appeared 90 times in 24 *SlPLATZ* genes, were the most prevalent *cis*-acting elements among the hormone response categories. MSA-like was present in the *SlPLATZ22* and *SlPLATZ24* genes. The promoter of *SlPLATZ8* has an RY element, which may control how the gene is expressed throughout the late stages of embryogenesis and seed formation. These findings suggest that *PLATZ* genes play a significant role in many aspects of life, including stress response and plant growth and development.

### 2.6. Interaction between PLATZ Proteins in Tomato

We mapped the interaction network and eliminated certain proteins with low values and missing annotations (Figure 7). The findings demonstrated that none of the proteins had any expected interactions with SlPLATZ17, SlPLATZ21, or SlPLATZ22. SlPLATZ1, SlPLATZ2, SlPLATZ18, SlPLATZ23, and SlPLATZ24 do not directly interact, but they work in concert with other related proteins to form 105 edges that control physiological processes. SlPLATZ23 interacted with the most proteins (12), followed by SlPLATZ1 and SlPLATZ2, which were each associated with 6 and 7 proteins, respectively, and SlPLATZ18 and SlPLATZ24, which each interacted with just one protein. These findings showed that ADH proteins were crucial for abiotic stress resistance, including resistance to waterlogging, low temperatures, drought, and salt damage [23]. HSP90 is crucial for the stage change from feeding to reproduction and floral development [24]. HSC80 is the principal heat shock protein generated by heat shock in yeast, mammals, and plants and is involved in particular developmental processes that are primarily linked to growth and development [25]. Temperature-sensitive circadian rhythms are linked to JA1, and the cell cycle, proliferation, apoptosis, and differentiation are connected to SGT1-1. The roles of other proteins in the interaction network diagram that appeared to have direct or indirect synergy with SlPLATZ proteins but lacked annotation are not known.

### 2.7. Tissue-Specific Expression Patterns of PLATZ Genes

Heat maps of tomato *SlPLATZ* gene-specific expression revealed that the levels of expression of various members of this gene family varied greatly in various tissues (Figure 8). Both *SlPLATZ17* and *SlPLATZ21* were significantly expressed in roots and to a lesser extent in cotyledons, demonstrating that these two genes are core members of the PLATZ family and are essential for tomato growth and development. *SpPLATZ7* and *SpPLATZ11* are highly expressed in roots and mature leaves at 6 weeks, showing that they are the core members of the PLATZ family and are essential for the growth and development of tomatoes (Appendix A). We discovered that *SlPLATZ21* and *SpPLATZ11*, *SlPLATZ17*, and *SpPLATZ7* are on the same branch of the phylogenetic tree and that all four genes are in group VI. This further emphasizes the significance of the tomato PLATZ family genes SlPLATZ17 and SlPLATZ21. *SlPLATZ21* and *SlPLATZ22* are predicted to be a pair of fragment replicates, but their expression patterns are not identical. *SlPLATZ21* was primarily expressed in roots, but *SlPLATZ22* was highly expressed at 0 DPA. Although it is projected that *SpPLATZ8* and *SpPLATZ13* are a pair of fragment clones, their expression profiles differ from one another. *SpPLATZ13* is expressed at a lower level in all tissues than *SpPLATZ8*, which is mostly expressed in roots. *SpPLATZ12* expression was lower than *SpPLATZ9* expression in each tissue, while *SpPLATZ9* expression in the bud was higher and *SpPLATZ12* expression in the mature fruit and 6-week small leaves was higher. (Appendix A). This result suggests that some *SlPLATZ* genes with close phylogenetic relationships may also exhibit different expression patterns. It was hypothesized that *SlPLATZ22* in Group VI and *SlPLATZ23* in Group IV, which were largely expressed at 0 DPA, would be involved in fruit growth and development. The importance of this gene was demonstrated by the fact that *SlPLATZ2* in Group II was primarily expressed in plant meristematic tissues, which further confirmed that this gene was common to monocotyledonous and dicotyledonous plants and that it was conserved.

### 2.8. Expression Profiles of SlPLATZ Genes

The relative expression of non-stressed tomato plants in roots, stems, and leaves was assessed using qRT-PCR with the primers shown in Table 1. As shown in Figure 9, *SlPLATZ2*, *SlPLATZ23*, and *SlPLATZ24* all showed the highest levels of expression in the root, demonstrating the significance of these three genes for root growth and development. The stem showed the highest expression of *SlPLATZ21* (1.120-fold, *p* = 0.125), followed by *SlPLATZ23* (0.585-fold, *p* < 0.0001), *SlPLATZ2* (0.278-fold, *p* < 0.0001), and *SlPLATZ24* (0.003-fold, *p* < 0.0001). The highest expression of *SlPLATZ21* (2.164-fold, *p* < 0.0001) was found in the leaves, suggesting that this gene primarily functions in leaves. *SlPLATZ23* (0.452-fold, *p* < 0.0001), *SlPLATZ2* (0.097-fold, *p* < 0.0001), and *SlPLATZ24* (0.004-fold, *p* < 0.0001) were the next most expressed genes. Most tissues showed high levels of *SlPLATZ21* expression, whereas stems and leaves showed relatively low levels of *SlPLATZ24* expression. The *SlPLATZ* gene may be implicated in the development of various tomato tissues, according to these crucial cues about expression variations and similarities in various tissues.

At the expanding period, mature green period, breaker period, and red ripening period under treatments of 10, 20, and 30 mg·L^−1^ NAA, 0.05, 0.1, and 0.2 mg·L^−1^ EBR, and 50, 100, and 150 µmol·L^−1^ MT, the relative expression of 11 representative genes in the tomato *PLATZ* gene family was examined using qRT-PCR (Figure 10). *SlPLATZ9* (314.929-fold) was highly expressed in red-ripening fruits sprayed with 20 mg·L^−1^ NAA, and the expression of NAA sprayed at ripening was much higher than in other periods. *SlPLATZ9* expression was higher in red-ripening fruits sprayed with BR, and the expression of BR sprayed at red ripening was much higher than in other periods. A change occurred until MT spraying, and *SlPLATZ9* expression was highest at 45.15-fold at the red ripening stage of fruits sprayed with 150 µmol·L^−1^ MT. Therefore, it is hypothesized that *SlPLATZ9* acts primarily at the red ripening stage of the fruit, which is consistent with the results obtained in tissue-specific expression. Except for *SlPLATZ21* (6.169-fold), which was more highly expressed in green ripening fruits sprayed with 0.1 mg·L^−1^ EBR and expanding fruits treated with 0.2 mg·L^−1^ EBR, *SlPLATZ21* (3.124-fold) was more highly expressed in expanding fruits sprayed with 20 mg·L^−1^ NAA. Compared to NAA spraying, *SlPLATZ21* expression was higher at the fruit expansion period with 50 µmol·L^−1^ MT (3.142-fold) and 100 µmol·L^−1^ MT (3.099-fold) spraying. Therefore, it is speculated that *SlPLATZ21* acted primarily during fruit expansion. *SlPLATZ23* (6.472-fold) was substantially expressed in mature green fruits sprayed with 30 mg·L^−1^ NAA, and the expression of NAA sprayed at mature green was significantly higher than at other times. *SlPLATZ23* expression was also higher than at other times in red-ripening fruits sprayed with EBR and significantly higher in mature green fruits sprayed with EBR and MT. Therefore, it is hypothesized that *SlPLATZ23* mostly affects fruits while they are in the green ripening stage. Consistent with our findings, we also observed increased *SlPLATZ23* expression at 0 DPA in tissue-specific expression.

To measure the transcript expression levels in the salt-stressed leaf tissues of cultivated tomatoes at 0, 0.5, 2, 4, 6, 8, and 12 h, qRT-PCR was used. The salt treatment results showed that all *SlPLATZ* genes were up-expressed in the leaf tissue of cultivated tomatoes (Figure 11). At 2 and 8 h, *SlPLATZ1* expression in leaf tissue was significantly elevated (13.699- and 16.135-fold), and the lowest expression occurred at 4 h (3.533-fold). The expression of *SlPLATZ21* in leaf tissue was significantly increased by 3.434- and 3.056-fold at 2 and 12 h, respectively, with the lowest expression of 1.377-fold observed at 8 h. *SlPLATZ1* expression was higher than *SlPLATZ21* expression for all salt stress treatment durations.

## 3. Discussion

Favorable conditions have been developed for the in-depth research of plant growth and development as well as resistance genes in recent years thanks to the expansion of genome sequencing. A useful research technique now is the analysis of gene families to investigate the structure, function, and evolution of genes. Studies related to *PLATZ* genes have been reported in higher model plant species such as Arabidopsis [17], maize [11], soybean [19], and rice [6]. Despite reports that the tomato *PLATZ* gene family has been identified. The distinction is that we discovered 24 *SlPLATZ* genes and examined them in terms of evolution. The expression of *SlPLATZ* was measured at different time points after salt stress, and the expression of fruits at four ripening stages was measured under NAA, BR, and MT conditions, respectively. Future research on tomato PLATZ will benefit from some of these consequences.

Cultivated tomato members of the PLATZ family were carefully identified in the present study, and their physicochemical characteristics, structural characteristics, evolutionary categorization, and functional expression were examined. Cultivated tomatoes, in contrast to most higher plants, have 24 members of the PLATZ family. This number is comparable to cabbage and higher than the numbers for Arabidopsis [17] and Ginkgo [26]. Compared to Wai’s [27] findings, which discovered 20 *SlPLATZ* genes, this is different. We found that Wai used a tomato assembly version other than Sl5.0 L., which may be the main reason for the disparity in the number of *PLATZ* genes. *PLATZ* members in wild tomatoes were primarily found on chromosomes 1, 2, 4, 6, 7, 8, and 10, but the PLATZ family was heterogeneously dispersed on chromosomes 1, 2, 3, 4, 6, 7, 8, and 12 in cultivated tomatoes. However, tomatoes from farmed and wild populations had the highest distribution of *PLATZ* genes on chromosome 2. Due to their locations on the chromosome, the *PLATZ* genes in tomatoes exhibit a mix of conservation and segregation. We note that *SlPLATZ11* is predicted to locate in mitochondria and that localization of transcription factors in mitochondria is uncommon. However, transcription factors have been found to have a dual location in the nucleus and other organelles, such as the mitochondria. Whirly1 provided the first instance of a protein that was simultaneously localized to the nucleus and an organelle within the same cell [28]. RNA and/or DNA-binding proteins (such as transcription factors and telomere binding proteins) make up the majority of the dual-targeting proteins (nuclear-organelle proteins) that are currently known. Such dual-targeted factors play a key role in nuclear versus cytosolic gene expression [29]. The 24 *SlPLATZ* genes identified in wild tomatoes and the 13 *SpPLATZ* genes identified in cultivated tomatoes were classified into four subfamilies. The exon-intron counts and conserved motifs of members of the same subfamily were similar, and members of various subfamilies had very different gene structures and conserved motifs. Even though we did not analyze the motifs of Arabidopsis and rice but rather those of tomato and the wild tomato Pennellii, Wai [27] discovered that motif1 is shared by the majority of *PLATZ* genes, which is similar to our findings. Intron loss controls intron evolution, and genetic mutations and selection affect intron evolution. Introns are lost after differentiation, and intron content is elevated during the early stages of gene amplification [30]. Generally, the number of introns within a subfamily is particular, and variations in the number of introns between various genes may result from insertion and deletion events, which may act as a driving force in the course of evolution. This force may also play a significant role in the diversity of gene structure and the complexity of gene function. Introns regulate the expression of genes either positively or negatively. Therefore, the functionality of *SlPLATZ* genes and their functional diversity may be impacted by conserved and variable gene architectures [31].

The growing complexity and refinement of morphological structures at various levels may be observed in the evolution of the biological world. Therefore, physiological functions have gradually expanded in specialization and efficacy. Because phylogeny is closely related to individual development, organisms always embody their ancestral characteristics early in their individual development before they embody their own more progressive characteristics. Phylogeny is the study of the origin and evolutionary history of biological groups, each of which has its own origins and developmental history, regardless of its size. Along with the organisms we included together, I left out Amborella (*Amborella trichopoda*), moss (*Physcomitrella patens*), and green algae (*Chlamydomonas reinhardtii*) concerning Wai’s [27], but with Pennellii included. The *PLATZ* genes differed significantly depending on where the plant was in its evolutionary history. According to the evolutionary tree, the evolution of the PLATZ family genome occurred concurrently with the diversification of its species. Monocotyledons and true dicotyledons are distributed in different branches. With a strong bias toward characteristics that are appropriate for human cultivation, such as high yield and large fruits, domestication is a process of human selection. Some *PLATZ* gene functions may be consistent with our domestication targets because domestication increased the number of *PLATZ* gene families in Solanaceae species. The monocotyledonous and dicotyledonous members of the phylogenetic tree were found in three subclades other than Group III, which indicated that *PLATZ* genes varied before monocotyledon-dicotyledonous divergence. The *PLATZ* genes were amplified during the differentiation of monocotyledons and dicotyledons because more genes were needed to perform more complex life activities. Duplicate genes in a phylogenetic tree belong to the same group, which provides the building blocks for the evolution of mechanistic novelty and makes it easier for new functions to evolve. While we discovered three gene pairs in cultivated tomatoes and four duplicated pairs in pennellii, Wai [27] discovered a total of seven duplicated gene pairs in the tomato *PLATZ* gene family. A ratio of non-synonymous substitutions (Ka) to synonymous substitutions (Ks) greater than 1 is widely thought to signal positive selection for faster evolution, and Ka/Ks < 1 shows that gene duplication suffers from purifying selection [32]. The three homologous gene pairs in the cultivated tomato similarly had Ka/Ks ratios that were <1, and all 20 *BrPLATZ* paralogous gene pairs were <1 [33]. In Wai’s study, These duplicated gene pairs may have diverged in the last 1–31 million years (Mya) [27], while our study found that cultivated tomato *SlPLATZ* gene pairs and *spPLATZ* homologous gene pairs in pennellii diverged at 26.600 and 31.767 Mya and 20.813 to 29.376 Mya, respectively. This may be related to the fact that we are using a different calculation method. Because there is more selection pressure on these duplicated gene pairs, evolution does not result in any appreciable functional changes between these pairs.

*Cis*-acting regulatory elements also regulate tissue-specific or stress-responsive expression patterns of genes. These elements are significant molecular switches that govern vast networks of genes involved in a variety of biological activities, including stress responses and developmental processes [34]. The promoter regions of most *SlPLATZ* genes contained several stress response cis-elements, which suggests that these genes play a role in abiotic stress and environmental adaptation. Plant hormones function as intracellular messengers in reaction to stress and combine several signal transduction pathways. Growth hormone, methyl jasmonate, gibberellin, and other phytohormone-responsive cis-elements are only a few of the phytohormone response elements found in the promoter regions of all *SlPLATZ* genes, which support the tolerance-related activities of *SlPLATZ* genes.

Protein interactions may connect PLATZ proteins to other known-function proteins, which advances our understanding of the biological roles of these proteins. Most PLATZ proteins may be involved in the cell cycle, proliferation, apoptosis, differentiation, and other processes, and the responses to abiotic stress, according to protein interactions. ADH [35] is produced under hypoxic conditions in maize and rice [23], and it is directly related to the organism’s degree of hypoxia. ADH encodes alcohol dehydrogenase, which is a key enzyme in anaerobic metabolism. Some of the key biological activities regulated by SGT1 include kinetochore assembly, ubiquitination, the activation of the cyclic AMP pathway, and the stability of Polo kinase [36]. SGT1 controls the growth hormone response, and it is an HSP90 co-chaperone. The growth hormone-responsive phenotype is buffered by the ability of HSP90 to hide point mutations in the growth hormone receptor TIR1 [37]. During normal tomato growth, HSC80 is preferentially expressed in the tips of the branches and roots [25].

The varying levels of gene expression in various tissues may be used to predict the diversity of function. The varying levels of *SlPLATZ* gene expression in diverse tissues suggest that functional diversity occurred during evolution, and these genes may serve a variety of regulatory functions during tomato growth and development. Most *SlPLATZ17*/*18*/*21*/*24* expression was found in roots, which indicated that these genes may be involved in root development or nutrient and water uptake. Because *SlPLATZ22*/*23* were expressed primarily at 0 DPA, these genes most likely collaborated with other fruit-specific genes to influence fruit development. Additional evidence that *SlPLATZ2* encouraged differentiation in tomatoes was based on its higher expression in meristematic tissues compared to other tissues. While we downloaded the *SlPLATZ* gene expression levels of various tomato tissues from the Tomato Functional Genomics Database (http://ted.bti.cornell.edu) (accessed on 18 September 2022), Wai [27] used RT-qPCR to investigate the gene expression patterns of several tomato tissues of the cultivar Ailsa Craig. *PLATZ* genes play a role in the growth and development of other plant species. For example, *PLATZ1* gene expression is more prevalent in the apical shoots and root tips of pea plants than in the mature leaf, stem, and root tissues [4]. *AtPLATZ7* regulates the size of the root meristem via ROS signaling [10]. Amyloplasts of maize endosperm selectively express *ZmPLATZ12* (Fl3) [11]. *GbPLATZ9* is strongly expressed in roots and seeds during late seed development. *AtORE15* is implicated in the control of leaf growth and the prevention of senescence, and it is expressed in young leaves [7]. *PLATZ18* and *PLATZ24* are two paralogous homologous gene pairs that have been duplicated, which suggests that these genes play a role even after replication and are involved in the control of tissue formation. The other two duplicated paralogous homologous gene pairs appear to have distinct functions in tomato development based on their varied expression patterns. In summary, we suggest that the *SlPLATZ* gene family, specifically the involvement of its members in abiotic stressors and fruit development, plays a major role in tomato development.

The expression levels of many genes varied as a result of tissue specificity. Under salt stress or after receiving various hormone treatments, this variation was expressed. For the demonstration, we chose a few genes that had high levels of gene expression. Roots are the main organ responsible for the absorption of water and minerals. In this study, *SlPLATZ23* was found to be highly expressed in the root system, indicating that it plays a significant role in root development and nutrient and water uptake. *SlPLATZ21* is abundantly expressed in plant leaves, indicating that it plays an important role in plant growth and development. The role of *PLATZ* genes in plant growth and development has been demonstrated in other species. In peas, for instance, the *PLATZ1* gene was expressed at higher levels in the root tips and apical shoots compared to mature leaf, stem, and root tissues [4]. The PLATZ transcription factor *AtORE15* gene is expressed in young leaves, and the gene is involved in regulating leaf growth and suppressing leaf senescence [7].

In resisting external stresses, plants often regulate their growth and resilience through hormone secretion, distribution, or signaling. About 15% of all plant transcription factors are zinc finger transcription factors, which are relatively large transcription factors that control the expression of many genes in response to abiotic conditions such as low temperature, salt, drought, osmotic stress, and oxidative stress [38,39]. *AtPLATZ1* and *AtPLATZ12* are the main nodes that positively regulate drought tolerance in Arabidopsis seeds and clonal tissues [17]. *GhPLATZ1* gene expression was induced by adversity and exogenous hormone stress. Ectopic expression of *GhPLATZ1* in Arabidopsis enhances plant resistance to osmotic stress, ABA, and PAC [6]. Similarly, the RNA expression of *GmPLATZ1* (Glycine max PLATZ1) was significantly increased when exogenous ABA was applied in soybean, and the leaf mRNA level of *GmPLATZ1* was steadily increased after 24 h of drought stress [19]. *SlPLATZ1* and *SlPLATZ21* expression considerably increased after 12 h of salt stress treatment in the current study; as a result, these genes may be crucial candidates for tomato’s salt stress response.

Transcription factors are involved in hormone signaling pathways [40]. Different endosperm treatments revealed that *ZmPLATZ2* was negatively controlled by ABA and sucrose and positively regulated by glucose. In soybean, *GmPLATZ1* is a stress-inducible PLATZ family protein that is overexpressed in Arabidopsis and responds strongly to both mannitol and abscisic acid [19]. In the present study, we found that *SlPLATZ21* and *SlPLATZ23* were significantly upregulated in tomato fruits at the MG stage after NAA and BR treatment and at the EP stage after MT treatment. However, further studies are needed to determine the mechanisms of NAA, BR, and MT induction of *SlPLATZ21* and *SlPLATZ23* and their roles in NAA, BR, and MT. Wai [27] investigated the effects of drought, salt stress, low temperatures, heat stress, and ABA on tomato Ailsa Craig. Our study, however, investigated the gene expression profiles of “M82” under various salt treatments and the expression patterns of cherry tomatoes “Jingfan Pink Star No. 1” at various stages of ripeness after spraying with various concentrations of EBR, NAA, and MT. This study laid the foundation for the establishment of the *SlPLATZ* gene family, but its role in different adversities still needs to be further validated by further experiments.

## 4. Materials and Methods

### 4.1. Plant Materials, Growth Conditions, and Treatment

The tomato cultivar “M82” was provided by the Institute of Horticultural Crops, Xinjiang Academy of Agriculture Sciences. The plants were grown in the Key Laboratory of Genome Research and Genetic Improvement of Xinjiang Specialty Fruits and Vegetables under conditions that included 16 h of light and 8 h of darkness, a temperature of 22 °C, 100 µmol·m^−2^·s^−1^ light intensity, and 60% relative humidity. Three tissue-specific samples were taken from tomato tissues, each consisting of material from three untreated 8-week-old plants. These three sample pools included leaves, stems, and roots. During the four-leaf stage, uniformly growing tomato seedlings were chosen, treated with 200 mmol·L^−1^ NaCl for stress, and sampled at 0, 0.5, 2, 4, 6, 8, and 12 h. Three biological replicas of each treatment were established, snap-frozen in liquid nitrogen for storage, and then stored at −80 °C.

The same supplier provided the “Jingfan Pink Star No.1” cherry tomato, which was grown in our facilities. The cherry tomato fruit was exposed to different concentrations of 2,4 epibrassinolide (EBR, concentration gradient of 0.05, 0.1, and 0.2 mg·L^−1^), naphthenic acid, NAA (concentration gradient of 10, 20, and 30 mg·L^−1^), melatonin (MT, concentration gradient of 50, 100, and 150 µmol·L^−1^), and control (CK) at the expanding period, mature green period, breaker period, and red ripening period. A total of 10 treatments were performed. The unfolding agent was Tween-80, and the solvent was 98% ethanol, which was dissolved and diluted to 0.1% (*v*/*v*). The experimental fruit was chosen at the mature green, breaker, and red ripening stages, and by the time it reached red ripening, its fruit had also been sprayed with comparable concentrations of exogenous NAA, EBR, and MT in the first three phases (cumulatively). Three biological duplicates were established for each treatment, and the sprayed fruits were sampled sequentially by zone at the expanding, mature green, breaker, and red ripening periods. The samples were flash-frozen in liquid nitrogen and stored at −80 °C.

### 4.2. Identification of PLATZ Gene Families

The Solanaceae genome database (http://solomics.agis.org.cn/tomato/ (accessed on 18 September 2022)) was used to gather reference genomes of tobacco (*Nicotiana tabacum* L., v4.5), eggplant (*Solanum melongena* L., v4.1), pennellii tomatoes (*Solanum pennellii*, v2.0), and cultivated tomatoes (*Solanum lycopersicum* L., Sl5.0). The Arabidopsis (*Arabidopsis thaliana*, TAIR10.) reference genome was gathered from the Arabidopsis Information Resource website (http://www.arabidopsis.org/ (accessed on 18 September 2022)). The Ensembl Plants database, located at http://plants.ensembl.org/index.html (accessed on 4 October 2022)), was used to compile reference genomes for the following plants: pepper (*Capsicum* L., v2.0), potato (*Solanum tuberosum*, v3.0), rice (*Oryza sativa*, IRGSP-1.0), maize (*Zea mays*, Zm5.0), wheat (*Triticum aestivum* L., IWGSC), sorghum (*Sorghum bicolor*,v3.56), soybean (*Glycine max*, v2.1), and cotton (*Gossypium_raimondii*,v6). A Markov model of the PLATZ protein’s conserved structural domain (PF04640) was acquired from Pfam (https://pfam.xfam.org/ (accessed on 18 September 2022)). The complete genome sequences of tomato M82 and Pennellii were searched using the hmmsearch (E-value: 1 × 10^−5^) algorithm to identify putative *PLATZ* genes [41]. The SMART (https://smart.embl.de/ (accessed on 19 October 2022)), NCBI-CDD (https://www.ncbi.nlm.nih.gov/cdd (accessed on 19 October 2022)), and Pfam databases (https://pfam.xfam.org/ (accessed on 18 September 2022)) all manually validate the structural domains of candidate genes. To have a more comprehensive understanding of the *PLATZ* gene family in Solanaceae, we analyzed Sl and Sp.

The Extasy online database ProtParam program (http://www.expasy.org/protparam/ (accessed on 19 October 2022)) was used to predict and examine the physicochemical properties of PLATZ proteins, including amino acid number [42], isoelectric point, lipid index, and hydrophilicity. WoLF PSORT (https://wolfpsort.hgc.jp/ (accessed on 19 October 2022)) was used to predict protein subcellular localization [43].

### 4.3. Conserved Structural Domains and Cis-Regulatory Elements of the PLATZ Gene

The MEME program [44] was used to find the conserved motifs in PLATZ proteins, with the number of motifs set to 10, the lowest motif length set to 6, and the largest motif length set to 100. Tbtools was used to visualize gene structure and motifs. The visualization of the gene structure and motif was completed using Tbtools [45]. Using the tomato genome annotation file, the entire 2000 bp upstream and downstream sequences of the tomato PLATZ gene were extracted and used as candidate promoter sequences using PlantCare (http://bioinformatics.psb.ugent.be/webtools/plantcare/html/ (accessed on 19 October 2022)) website [46], which was analyzed for *cis*-acting elements and mapped using Tbtools software [45].

### 4.4. Selection Pressure Analysis, Interspecies Collinearity, and Phylogenetic Analysis of the PLATZ Gene

The Ka/Ks Calculator was used to determine the ratio of the *PLATZ* gene synonymous with the nonsynonymous substitution rate (Ka/Ks) [47]. The *PLATZ* gene covariance between and within species was calculated using McscanX [48]. Circos was used to demonstrate covariance between genes. The combined dataset was analyzed with maximum likelihood inference using Iqtree [49]. The ModelFinder model with a bootstrap value of 1000 was used to further visualize and embellish the phylogenetic tree with the help of the iTOL online tool [50].

### 4.5. Interaction Network and Expression Analysis

The protein-protein interaction relationship was predicted by the STRING online website (https://string-db.org/ (accessed on 3 November 2022)), using the Cytoscape program to display an interaction network [51]. Expression profiles of *S. lycopersicum* at various tissues and developmental stages were downloaded from the RNA-seq database of the Tomato Functional Genome Database (TFGD, http://ted.bti.cornell.edu/ (accessed on 2 November 2022)) [52], including expression data for anthesis flowers (0DPA), 10 days post-anthesis fruit (10 DPA1), 10 days post anthesis fruit 2 (10 DPA2), 20 days post-anthesis fruit (20DPA), ripening fruit (33DPA), cotyledons (COTYL), hypocotyl (HYPO), vegitative meristems (MERI), mature leaves (ML), whole root (ROOT), young flower buds (YFB), and young leaves (YL). The expression profiles for the different tissues and developmental stages of *S. pennellii* were extracted from Appendix A attached to the article by Bolger et al. [53]. including the FPKM values of bud, flower, immature (IF), mature fruit (MF), 6-week, small leaves (6 wk, SL), 6-week, mature leaves (6 wk, ML), 6-week, Meristem (6 wk, M), 6-week, stem (6 wk, S), pollen, and root (SR). TBtools was used to perform a heat map analysis on their expression profiles.

### 4.6. Expression Profile of SlPLATZ

The Tiangen Biochemical Technology (Beijing) Co.’s RNA-prep Pure Plant Kit (DP441) was used to extract total RNA. The abm All-In-One 5× RT Master Mix Reverse Transcription Kit was used to reverse transcribe the cDNA to produce cDNA. The qRT-PCR primers for SlPLATZ family members were created using DNAMAN6 software. Table 1 lists the specific primers for the *SlPLATZ* genes. The following ingredients were included in the 20 μL qRT-PCR reaction system: 10 μL of the 2× ChamQ Universal SYBR qPCR Master Mix (Novozymes), 8.2 μL of ddH_2_O, 0.4 μL of upstream and downstream specific primers, and 1 μL of cDNA template. The reaction program was 120 s of pre-denaturation at 94 °C, 5 s of denaturation at 94 °C, 15 s of annealing time, and 10 s of extension at 72 °C for 45 cycles using a Roche Light Cycler 96 real-time quantitative fluorescence PCR apparatus. The internal reference gene was tomato actin. Three biological replicates and the 2^-ΔΔCT^ method were used to calculate the relative expression, which was the difference between the relative values of the treatment and control groups. The CT of each treatment was subtracted from the CT of actin to obtain the ΔCT of each treatment. The mean of the control ΔCT was then computed, and the ΔCT for each treatment was subtracted from the ΔCT! to obtain the ΔΔCT for each treatment. The relative expression is then calculated using Equation 2^−ΔΔCT^. To examine if the data were normally distributed, we utilized the GraphPad Prism 9.0 program [54]. To ascertain whether variations in expression were significant, a paired one-way ANOVA was carried out.

## 5. Conclusions

The 24 *SLPLATZ* genes that were found in this investigation were grouped into four subfamilies. From an evolutionary perspective, the entire tomato gene family of Solanaceae has not contracted or expanded from its current status, and the evolution of *PLATZ* family genes has not been affected by the separation of cultivated tomato from potato and cultivated tomato from pennellii. Among them, *SlPLATZ17*, *SlPLATZ21*, and *SlPLATZ22* underwent quantitative amplification rather than functional differentiation. *SlPLATZ21* acts mainly in the leaves. The expression of *SlPLATZ21* in leaf tissue was significantly increased after 2 and 12 h of salt stress. It is hypothesized that *SlPLATZ21* plays an important role in the growth and development of tomatoes under salt stress. *SlPLATZ23* was highly expressed in mature green fruits sprayed with NAA, as well as being much higher in mature green fruits sprayed with MT and mature green fruits sprayed with EBR than at other times. Thus, we proposed that *SlPLATZ23* mostly affects fruits at the stage of green ripening. This study will lay the foundation for a deeper understanding of the evolutionary mechanisms and functional properties of the *PLATZ* family in tomatoes.

## Figures and Tables

**Figure 1 plants-12-02632-f001:**
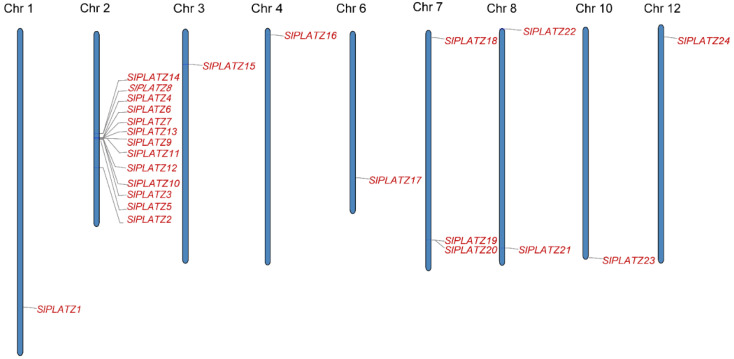
Location of the *SlPLATZ* genes on the chromosomes of Heinz tomatoes.

**Figure 2 plants-12-02632-f002:**
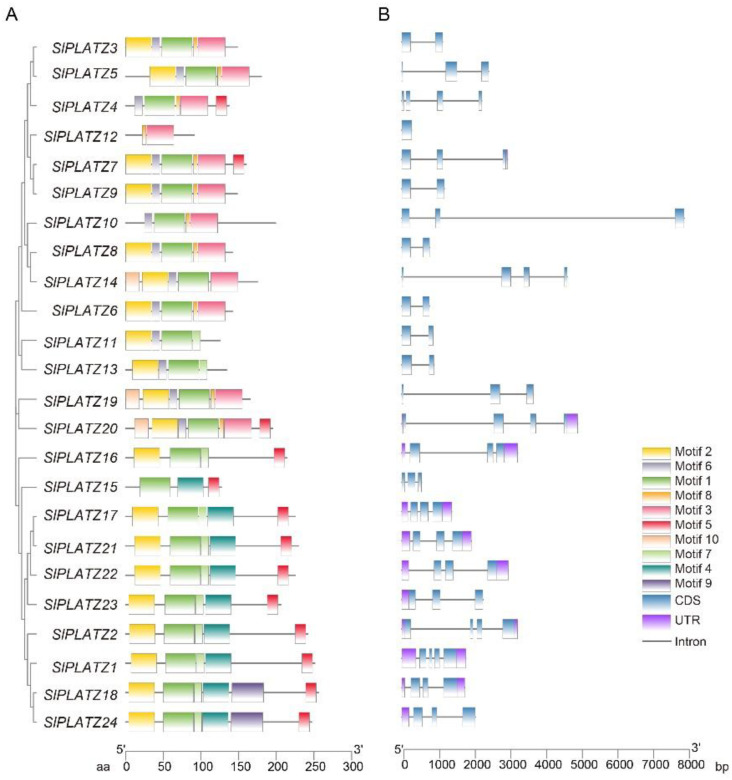
Analysis of motif structure, gene structure, and phylogenetic analysis of SlPLATZ. (**A**) Ten amino acid motifs in the SlPLATZ protein are indicated by a colored box. (**B**) The blue box indicates exons, the purple box indicates UTRs, and the gray line indicates introns.

**Figure 3 plants-12-02632-f003:**
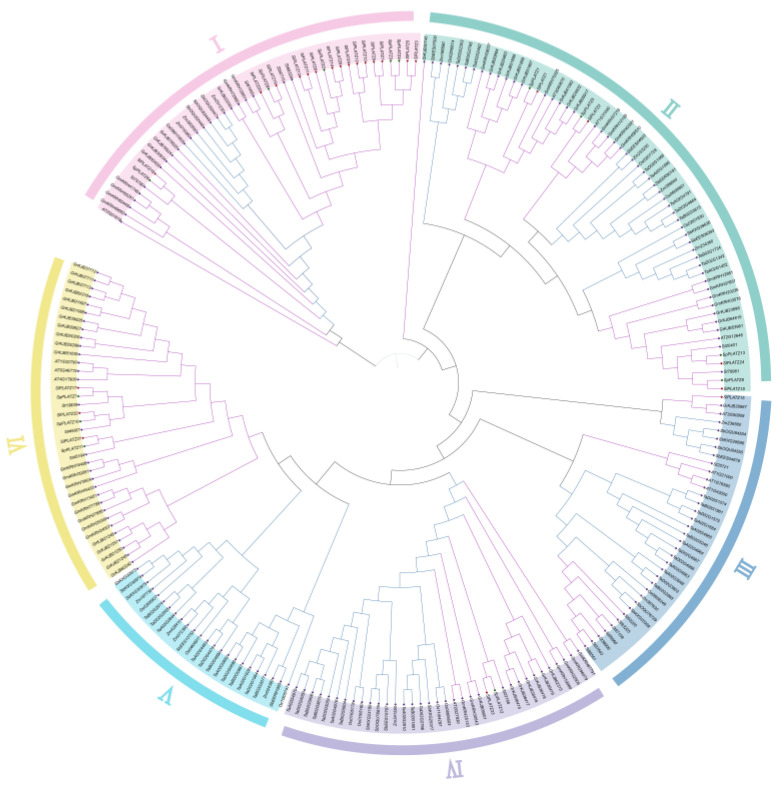
Phylogenetic tree of PLATZ family members. The red rhombus indicates the cultivated tomato, whereas the green rhombus identifies the pennellii tomato. The blue line connects monocotyledons, while the purple line connects dicotyledons.

**Figure 4 plants-12-02632-f004:**
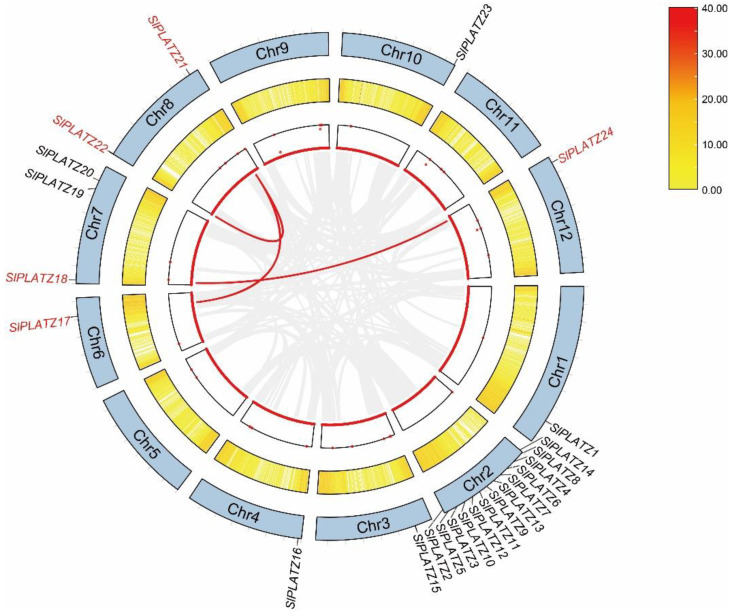
Gene duplication events in the genome, with the circles from outer to inner representing the chromosomes of cultivated tomatoes, the gene density, and the N ratio presented in red dots, and the ends of the red lines representing direct homologous *SlPLATZ* genes. The legend represents the values of the gene density, With red representing high levels and yellow representing low levels.

**Figure 5 plants-12-02632-f005:**
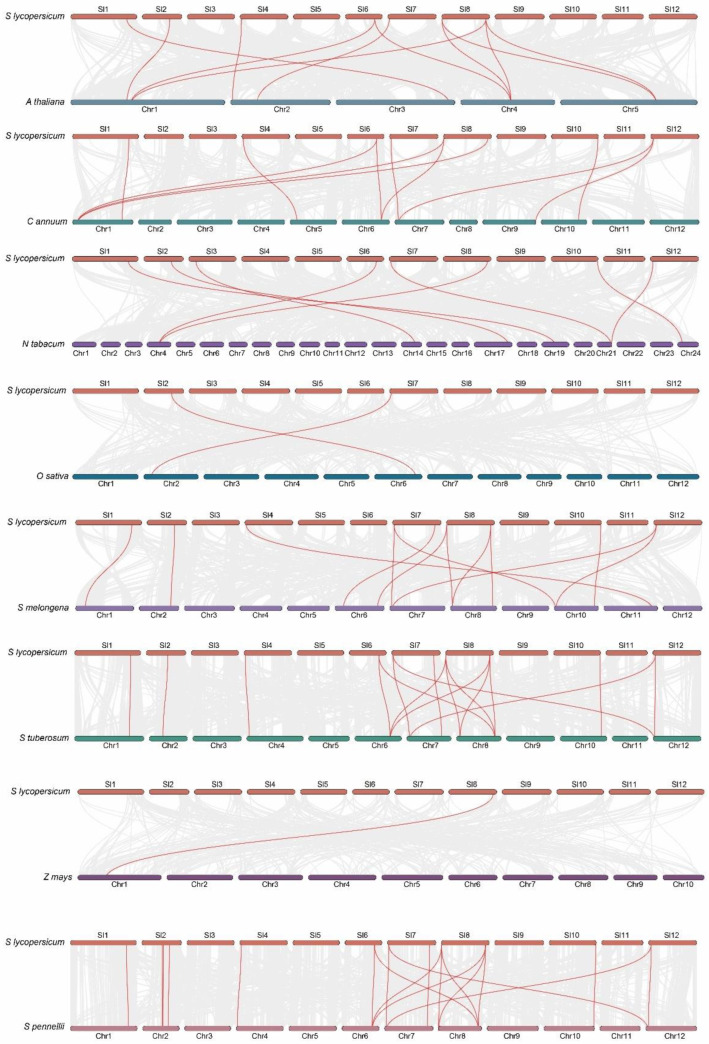
The collinearity diagram of *PLATZ* genes. Red lines highlight the homologous gene pairs of *SlPLATZ* genes, and gray lines represent genome-wide collinear gene pairs.

**Figure 6 plants-12-02632-f006:**
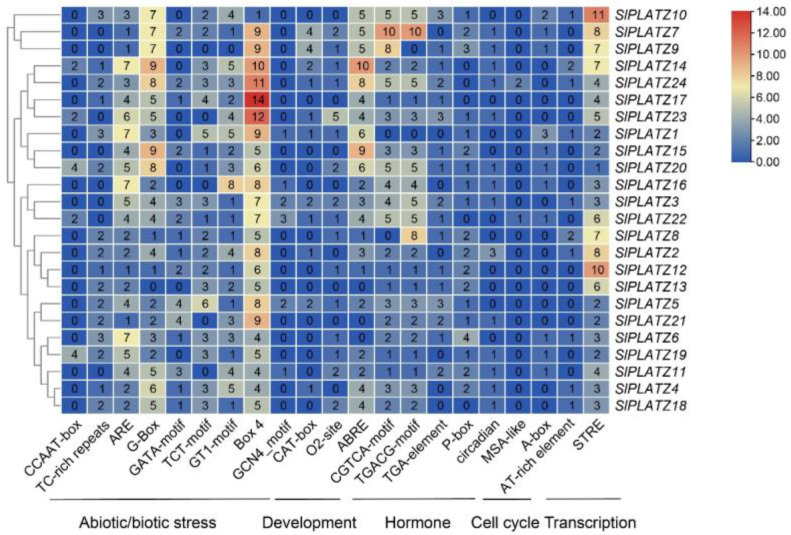
The number of *cis*-acting elements contained in the promoter of the *SlPLATZ* genes. The number of *cis*-acting elements involved in different regulatory pathways varies. Darker red colors indicate higher numbers of *cis*-acting elements. Darker blue colors indicate lower numbers of *cis*-acting elements.

**Figure 7 plants-12-02632-f007:**
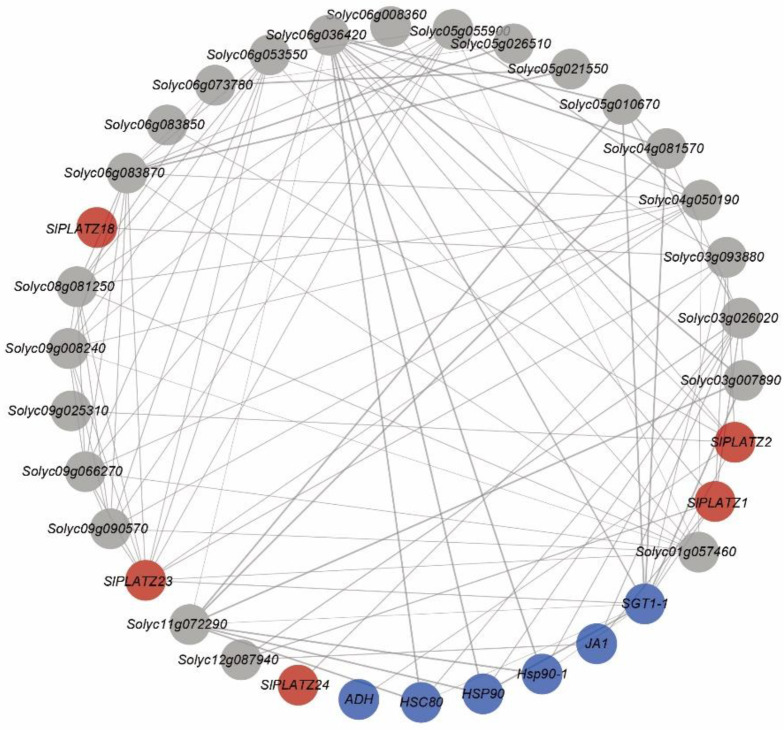
An interaction network of PLATZ genes in tomato. Each node is a protein, and each edge represents the presence of interactions. The size of the node represents the number of interactions. The thickness of the edge represents the value of the combined score. Red nodes represent PLATZ proteins, blue nodes represent stress-related proteins, and gray nodes represent proteins lacking annotation.

**Figure 8 plants-12-02632-f008:**
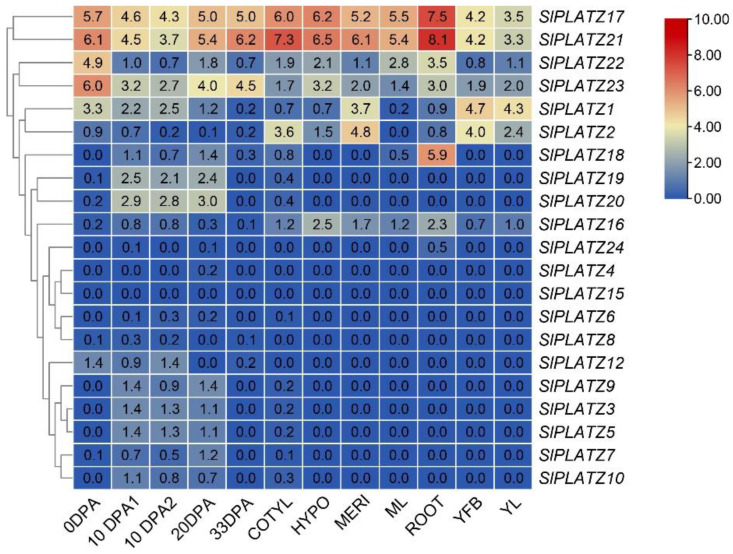
Heat map of tissue-specific expression of *PLATZ* genes in tomato. The color bar represents the log^2^ expression values, with red representing high expression levels and blue representing low expression levels. The gene name is shown on the right side. LA1589 20-day post-anthesis fruit (20 DPA), LA1589 10-day post-anthesis fruit (10 DPA1), LA1589 10-day post-anthesis fruit 2 (10 DPA2), LA1589 root (ROOT), LA1589 anthesis flowers (0 DPA), LA1589 vegetative meristems (MERI), LA1589 young flower buds (YFB), LA1589 young leaves (YL), LA1589 hypocotyl (HYPO), LA1589 cotyledons (COTYL), LA1589 ripening fruit (33 DPA), and LA1589 mature leaves (ML).

**Figure 9 plants-12-02632-f009:**
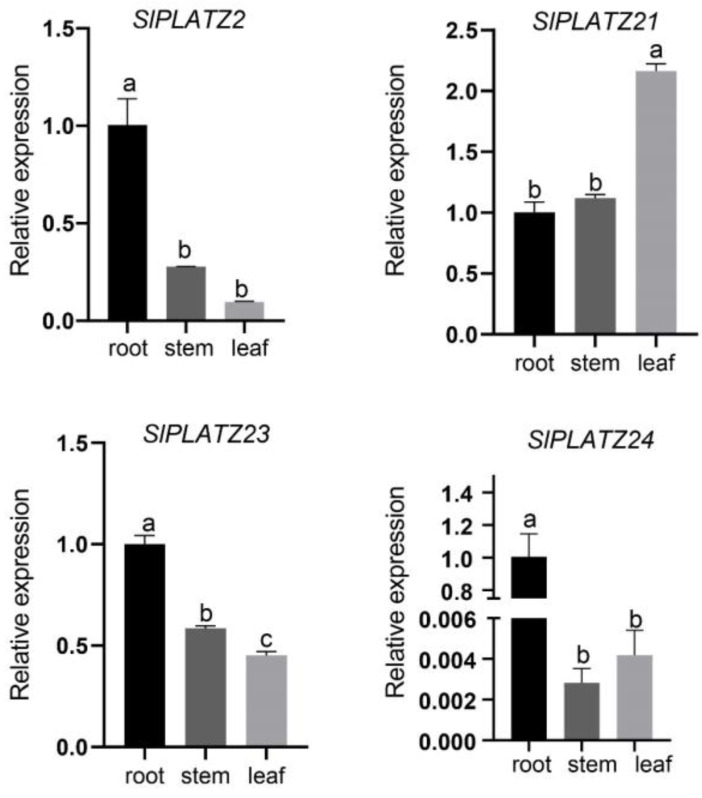
Relative expression of *SlPLATZ2*, *SlPLATZ21*, *SlPLATZ23*, and *SlPLATZ24* in roots, stems, and leaves. *p* < 0.0001. The standard is the largest average marked with a, and the following unmarked averages are marked with the letter a where it is not significant until a certain average that differs significantly from it is marked with the letter b. The largest average that begins with the letter b should then be used as a benchmark for comparison with the remaining averages. Continue to append the letter b to any non-significant variances. Until they differ sufficiently, at which point they are designated with the letter c.

**Figure 10 plants-12-02632-f010:**
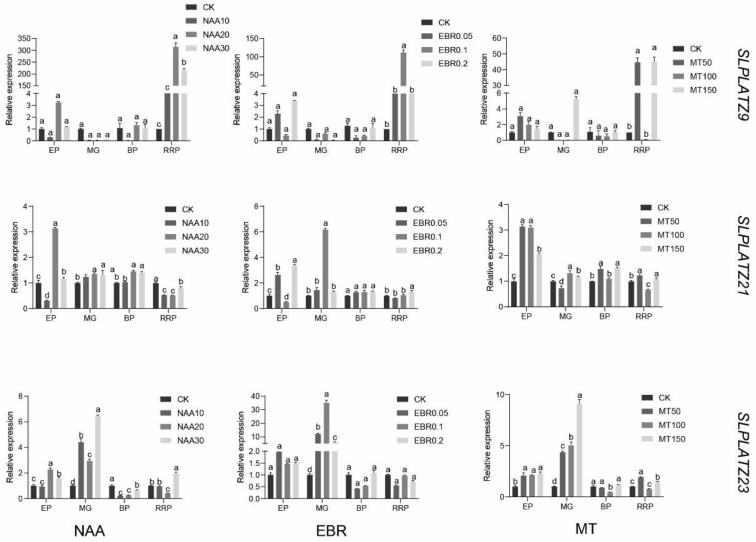
Expression analysis of the *SlPLATZ* genes under hormone stress during fruit development. The error bar shows the standard deviation of three biological repeats. EP—Expending period; MG—Mature green period; BP—breaker period; RRP—Red ripening period; CK—control; NAA—naphthenic acid; EBR—2,4 epibrassinolide; MT—melatonin. *p* < 0.0001. The standard is the largest average marked with a, and the following unmarked averages are marked with the letter a where it is not significant until a certain average that differs significantly from it is marked with the letter b. The largest average that begins with the letter b should then be used as a benchmark for comparison with the remaining averages. Continue to append the letter b to any non-significant variances. Until they differ sufficiently, at which point they are designated with the letter c. The largest average that begins with the letter c should then be used as a benchmark for comparison with the remaining averages. Continue to append the letter c to any non-significant variances. Until they differ sufficiently, at which point they are designated with the letter d.

**Figure 11 plants-12-02632-f011:**
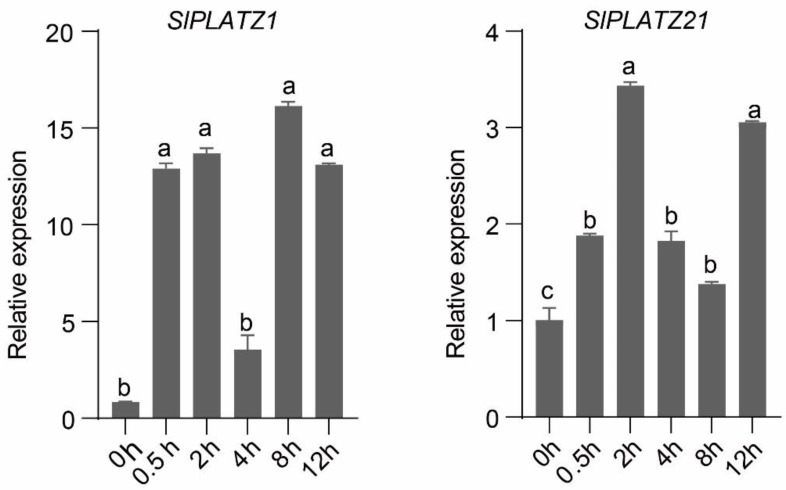
Analysis of salt stress expression in *SlPLATZ1* and *SlPLATZ21* leaves. *p* < 0.0001. The standard is the largest average marked with a, and the following unmarked averages are marked with the letter a where it is not significant until a certain average that differs significantly from it is marked with the letter b. The largest average that begins with the letter b should then be used as a benchmark for comparison with the remaining averages. Continue to append the letter b to any non-significant variances. Until they differ sufficiently, at which point they are designated with the letter c.

**Table 1 plants-12-02632-t001:** Primer sequences for qRT-PCR experiments.

Gene Name	Forward Primer (5′→3′)	Reverse Primer (5′→3′)
*SIPLATZ1*	CGGGTTTAGACGATGGTCAA	GTTTCTTCCTTACCACCTCTGTTG
*SIPLATZ2*	TGGAAGAGATGATGAAACCTGC	GAGGATGTGAGTGATGTTGTGG
*SIPLATZ4*	GCCACATTCTGGGTCTGTCT	CCACATTCTGGGTCTGTCTCAT
*SIPLATZ5*	ATTGCTTTGAATCCTCTGCCAC	TGAAACGGAGGGAGAGAAATG
*SIPLATZ12*	GCTTTGAATCCTCTGCCACA	CATCGCCAAACAACCACT
*SIPLATZ14*	CTTTGAATCCTCTGCCACATTG	CTTCGTCTCAACAATCTTTCCC
*SIPLATZ19*	TTGCGTTGCTACAGACGAAC	TTATGTGGCAGCGGATTCA
*SIPLATZ20*	ACAGACAAACACAATGGGCA	GCAGCGGATTCAAAGCAAC
*SIPLATZ21*	GGTTGAAGCCATTGTTGAAGG	CGATGACCTCCGAATCTGAAT
*SIPLATZ23*	TGCCGCTCATCTAAACACAA	TGCCACTGCTCTTTGGTTG
*SIPLATZ24*	GCTAAAGCACCACAAGGACCT	ATGGCTCGTCATTGTCCACG
*Actin*	CAGGGTGTTCTTCAGGAGCAA	GGTGTTATGGTCGGAATGGG

## Data Availability

Data will be made available upon request.

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
