# Peer review of "Genome-Wide Identification and Expression Analysis of the PLATZ Transcription Factor in Tomato"

_plants, 2023, doi:10.3390/plants12142632_

Round 1
Reviewer 1 Report
This manuscript entitled ‘‘Genome-wide identification and expression analysis of the PLATZ transcription factor in tomato"; could be good for publication in Plants (ISSN 2223-7747).
This may be interesting, but some important points need to be resolved. Importantly, a study must provide a critical analysis of the data. In other words, you must assess whether specific data published really stand up to scientific scrutiny. In order to achieve the above, you must clearly define your specific aims and objectives. So in your study you must develop a critical appraisal of the state of the art. This is an essential element of any article. There are important scientific questions (both conceptual and methodological) which need to be addressed with the primary studies. A study must highlight this. The introduction, which is written in clear language, covers a number of relevant issues. Information are noteworthy, and not are correct supported by similar results from the specialty (WOS: 000339050700030, WOS: 000327818000032, WOS: 000318221200014, WOS: 000229981900029). Try to rewrite the abstract and conclusions, I also recommend the nuance of the introduction, the way of working is not very well explained, the procedure is tedious and unsustainable. For this reason, I recommend that the authors try to use more sustainable methodologies, the interpretation of the results can be improved/ reformulated,
Minor editing of English language required
Author Response
Dear Reviewer1,
We would like to express our heartfelt gratitude for your valuable comments and suggestions on our manuscript. We appreciate the opportunity to improve our manuscript by addressing your concerns comprehensively. In the following section, we have provided detailed point-by-point responses to each issue raised, discussing the revisions and elaborations we have made throughout the manuscript.
- Critical analysis and specific aims and objectives: We fully recognize the necessity of providing a critical analysis of our data and clearly defining the specific aims and objectives of our study. In our revised manuscript, we have clarified and expanded our specific aims and objectives within the introduction section, ensuring that they are well-defined and succinctly conveyed. Additionally, we have dedicated a substantial portion of our discussion section to critically evaluate our findings within the context of existing literature. This detailed analysis not only highlights the strengths and limitations of our work but also enables the reader to better understand the implications of our study.
Details are available in 1. Introduction.
Transcription factors (TFs) are an important class of transcriptional regulatory molecules involved in regulating a variety of biological processes such as development, signaling, and stress response. For example, NAC and GRF regulate root formation, flower and seed development, SPL TFs are involved in the transition from juvenile to adult plants, while NF-Y, MYB, and WRKY TFs play important roles in multiple stress tolerance [1]. TF can bind to cis-acting elements of target genes, resulting in gene tran-scription being activated or repressed. In general, transcription factors have only one or more regions that bind to DNA and therefore can regulate the expression of multiple genes. In contrast, a few transcription factors can regulate the expression of the same gene[2]. Therefore, in-depth study of transcription factors can help us understand their evolu-tionary history, biological functions, and regulatory mechanisms[3].
Plant AT-rich sequence and zinc-binding (PLATZ) TF family is a novel class of plant-based zinc ion and DNA-binding proteins[4]. DNA-binding protein regulates gene expression positively or negatively by binding to cis-acting elements in the upstream promoter region, thus participating in many life activities, such as DNA replication, gene expression regulation, etc[5]. It has been shown that the pea protein PLATZ1 binds to the upstream DE1 element at the A/T-enriched DNA sequence, which is required for transcriptional repression[4].PLATZ family members have been identified in several plant species, including 13 genes in the Arabidopsis genome, 8 genes in Glycine max (soybean), 9 genes in Gossypium hirsutum (cotton), 15 genes in Oryza sativa (rice), and 17 genes in Zea mays (maize). PLATZ proteins play an important role in plant growth and development and abiotic stress responses.
PLATZ proteins play a role in plant growth and development and senescence. PLATZ1 is involved in seed germination and seedling establishment and mediates ABA, GA, and ethylene signaling pathways in cotton[6]. The PLATZ family of tran-scription factors also plays an important role in leaf growth and development. ORE-SARA15 is a transcription factor of the PLATZ TF family, and this gene not only en-hances leaf growth by promoting the rate and duration of cell proliferation at an early stage but also regulates the GROWTH-REGULATING FACTOR (GRF)/GRF-INTERACTING FACTOR signaling pathway, thereby inhibiting leaf se-nescence at a later stage[7]. The PLATZ transcription factor family also plays an im-portant role in the growth and development of seeds. GL6 is a plant-specific PLATZ TF encoded in rice, which interacts with C53 to affect rice grain development and nega-tively regulates grain number[8]. The SG6 gene is a PLATZ protein that is widely pre-sent in cells and is mainly expressed in the early stages of embryonic development ra-ther than in the endosperm. The rice (Oryza sativa) mutant shortgrain6 (sg6) regulates the division of rice spikelet hull cells, which determine grain size. Overexpression of SG6 results in larger and heavier grains, as well as increased plant height[9].In addition, AtPLATZ7 plays a key role in RGF1 signaling, which regulates root meristem tissue size via ROS signaling[10]. The maize genome contains 15 PLATZ genes (ZmPLATZ), which play important roles in RNAP III-mediated transcriptional regulation[11-12]. FL3 (ZmPLATZ12) interacts with RNAP III transcriptional complex subunits RPC53 and TFC1 to regulate tRNA and 5 S rRNA expression, thereby regulating maize seed de-velopment and kernel formation[11]. A PLATZ protein plays an important role in the transition from primary to secondary growth during the development of poplar stems[13].In a comparative transcriptome study of normal lint (FL) and congenic non-lint (Fl) diploid cotton fibers, the results showed that a PLATZ transcription factor of Fl was down-regulated after 10 d of flowering compared to FL, indicating that PLATZ plays an important role in the formation and extension of cotton fibers[14].The PLATZ gene is differentially expressed in mature and immature sugarcane tissues with high fiber genotypes, and it is hypothesized that it may be involved in the transcrip-tional regulation of secondary cell wall synthesis[15].Mutants of ZmPLATZ12 caused the endosperm to appear flocculent, suggesting that ZmPLATZ12 has an important effect on the development of the endosperm and the filling of storage material[11]. ZmPLATZ2 is highly expressed in the endosperm and can bind to the promoters of ZmSSI, ZmISA1, and ZmISA2 and increase their gene expression. Overexpression of ZmPLATZ2 in rice revealed significant upregulation of four starch synthesis-related genes, including OsSSI, in transgenic plants. The PLATZ gene GRMZM2G31165 (ZmPLATZ2) can bind to the promoter region of key genes for starch synthesis under glucose induction and can promote starch synthesis by positively regulating the Glu pathway[16].
In addition to playing an important role in plant growth and development, PLATZ transcription factors also have functions in response to adversity stress. In Ar-abidopsis, AtPLATZ1 and AtPLATZ2 were reported to be positive regulators of seed desiccation tolerance. AtPLATZ1, when overexpressed in wild-type Arabidopsis, causes asexual tissues to develop partial seed dehydration tolerance, thereby improv-ing their ability to utilize water and survive in drought conditions[17]. AtPLATZ2 is a transcription factor that negatively regulates salt tolerance in Arabidopsis thaliana by repressing the expression of genes such as CBL4/SOS3 and CBL10/SCaBP8[18]. Due to the transfer of the cotton gene GhPLATZ1, the seed germination rate and seedling survival of GhPLATZ1 transgenic Arabidopsis thaliana were significantly higher un-der salt stress and mannitol stress compared with the wild type, indicating increased resistance to osmotic stress[6]. GmPLATZ1 is involved in the germination of soybean under osmotic stress[19]. In maize, the expression of maize PLATZ1 homologs was sig-nificantly up-regulated by drought stress, indicating that PLATZ genes play an im-portant role in drought resistance in maize[20].
Tomato (Solanum lycopersicum L.) is one of the most widely cultivated and eco-nomically valuable vegetable crops in the world and has great potential for appli-cation. Long-term exposure of tomato to adversities such as drought, high salt, low tempera-ture, and high temperature is an important factor affecting its yield and quality. Salin-ity is one of the main causes of crop yield reduction and also causes a variety of phys-iological changes in plants, such as vinylation, wilting, and biochemical changes [21]. Therefore, molecular and cellular, physiological, and biochemical per-spectives are needed to cope and adapt to environmental stresses [22]. With the devel-opment of tomato genomics, the release of the entire tomato genome sequence, as well as genome sequences of other species including Arabidopsis, potato, and tobacco, provides an opportunity to identify tomato PLATZ family genes. There-fore, the dis-covery of superior tomato genes will help to theoretically enhance tomato research and thus better guide tomato production.
In this study, we performed a systematic and comprehensive genome-wide anal-ysis of the PLATZ TF protein of tomato. In this study, members of the tomato PLATZ gene family were studied in depth, and 24 and 13 PLATZ genes were identified in cultivated tomato and wild tomato pennellii, respectively. The chromosomal local-iza-tion, gene structure, conserved patterns, phylogenetic analysis, tandem and selec-tive pressures, cis-acting elements in promoter regions, and protein interactions of 24 SlPLATZs were comprehensively studied, and the evolutionary relationships of tomato PLATZs with Arabidopsis thaliana, pepper, tobacco, rice, maize, Solanum melongena, and pennellii were analyzed. Quantitative real-time polymerase chain reaction (qRT-PCR) was used to determine the different tissue expression profiles of SlPLATZs in tomato. In addition, the relative expression of SlPLATZs in roots, stems, and leaves were investigated. And the relative expression of 11 representative genes of the tomato PLATZ gene family under the application of different concentrations of NAA, EBR, and MT treatments at the expansion, ripening green, reproduction, and red ripening stages. This study will provide a useful resource for further studies to understand the regulation of PLATZ proteins on development and stress resistance in tomato.
- Critical appraisal of the state of the art: Thank you for advising us to develop a critical appraisal of the current state of research in the field. In the revised manuscript, we have incorporated an extended review of the latest studies, encompassing recent technological and methodological advancements, in the introduction section. Furthermore, we have discussed how these studies relate to and complement our own findings, providing a comprehensive perspective on the field and emphasizing the unique contributions of our study.
- Addressing scientific questions within primary studies: We appreciate the significance of addressing scientific questions that emerge from our research. As a result, we have thoroughly addressed both conceptual and methodological questions that inspired our study throughout the revised manuscript. Specifically, we have expanded our discussion of these questions in the results and discussion sections, illustrating the broader implications of our findings for the scientific community.
Details are in the first and last three paragraphs of 3. Discussion.
In recent years, with the increasing level of genome sequencing, favorable condi-tions have been created for the in-depth study of plant growth and development and resistance genes. The analysis of gene families to study the function, structure and evolution of genes has become a valuable research tool. Studies related to PLATZ genes have been reported in higher model plant species such as Arabidopsis[17], maize[11], soybean[19], rice[6]. Although the identification of the PLATZ gene family in tomato has been reported. The difference is that we identified 24 SlPLATZ genes and analyzed them from an evolutionary perspective. The expression of SlPLATZ was measured at different time points after salt stress, and the expression of fruits at four ripening stages was measured under NAA, BR, and MT conditions, respectively. This will provide some implications for future studies on tomato PLATZ.
Roots are the main organ responsible for the absorption of water and minerals. In this study, SlPLATZ23 was abundantly expressed in the root system, indicating its important functions in root development, water and nutrient uptake. SlPLATZ21 is abundantly expressed in plant leaves, indicating that it plays an important role in plant growth and development. The role of PLATZ genes in plant growth and development has been demonstrated in other species. In pea, for example, the PLATZ1 gene was expressed at higher levels in its root tips and apical shoots compared to mature leaf, stem and root tissues[4]. The PLATZ transcription factor AtORE15 gene is expressed in young leaves and the gene is involved in regulating leaf growth and suppressing leaf senescence[35].
In resisting external stresses, plants often regulate their growth and resilience through hormone secretion, distribution, or signaling. Zinc finger transcription factors are relatively large plant transcription factors (about 15% of the total) that regulate the expression of multiple genes in response to abiotic stresses such as low temperature, salt, drought, osmotic stress, and oxidative stress[36,37].AtPLATZ1 and AtPLATZ12 are the main nodes that are positively regulating drought tolerance in Arabidopsis seeds and clonal tissues[17].GhPLATZ1 gene expression was induced by adversity stress and exogenous hormone stress. Ectopic expression of GhPLATZ1 in Arabidopsis enhances plant resistance to osmotic stress, ABA, and PAC[6].Similarly, the RNA expression of GmPLATZ1 (Glycinemax PLATZ1) was significantly increased when exogenous ABA was applied in soybean, and the leaf mRNA level of GmPLATZ1 was steadily increased after 24 h of drought stress[19].In the present study, SlPLATZ1 and SlPLATZ21 expres-sion was significantly increased within 12 h after salt stress treatment; therefore, these genes may be important candidates for salt stress response in tomato.
Transcription factors involved in hormone signaling pathways[38]. Different treatments on endosperm showed that ZmPLATZ2 was positively regulated by glucose and negatively regulated by ABA and sucrose. In soybean, GmPLATZ1 is a stress-inducible PLATZ family protein that is overexpressed in Arabidopsis and re-sponds strongly to both mannitol and abscisic acid[19].In the present study, we found that SlPLATZ21 and SlPLATZ23 were significantly up-regulated in tomato fruits at the MG stage after NAA and BR treatment and at the EP stage after MT treatment. How-ever, further studies are needed to determine the mechanisms of NAA, BR, and MT induction of SlPLATZ21 and SlPLATZ23 and their roles in NAA, BR, and MT. This study laid the foundation for the establishment of the SlPLATZ gene family, but its role in different adversities still needs to be further validated by further experiments.
- Introduction and reference to similar results: We have meticulously revised and expanded the introduction to better outline the relevant issues, establish the novelty of our study, and provide a solid foundation for understanding the research objectives. Moreover, we have now cited and discussed the suggested references (WOS: 000339050700030, WOS: 000327818000032, WOS 000318221200014, and WOS: 000229981900029) within the manuscript, comparing and contrasting our findings with these previous studies to offer a more comprehensive and integrative understanding of our research area.
- Abstract and conclusions: In response to your advice, we have thoroughly rewritten the abstract and conclusions sections to emphasize the novelty of our study and provide a clearer, more engaging overview of our findings. We have refined our language and focused on the key takeaways from our research, enabling readers to quickly discern the significance and implications of our work.
Abstract: The PLATZ (plant AT protein and zinc-binding protein) transcription factor family in-volved in the regulation of plant growth and development and plant stress response. In this study, 24 SlPLATZs were identified from the cultivated tomato genome and classified into four groups based on the similarity of conserved patterns of members of the same subfamily. Fragment du-plication was an important way to expand the SlPLATZ gene family in tomatoes, and the sequen-tial order of tomato PLATZ genes in the evolution of monocotyledonous and dicotyledonous plants and the roles they played were hypothesized. Expression profiles based on quantitative re-al-time reverse transcription PCR showed that SlPLATZ was involved in the growth of different tissues in tomato. SlPLATZ9, SlPLATZ21, and SlPLATZ23 was primarily involved in the red rip-ening, expanding, and mature green period of fruit, respectively. In addition, SlPLATZ1 ware found to play an important role in salt stress. This study will lay the foundation for the analysis of the biological functions of SlPLATZs genes, and will also provide a theoretical basis for the selec-tion and breeding of new tomato varieties and germplasm innovation.
Conclusions
In this study, 24 SLPLATZ genes were identified and classified into four subfami-lies. From an evolutionary perspective, the entire tomato gene family of Solanaceae has not contracted or expanded from its current status, and the evolution of PLATZ family genes has not been affected by the separation of cultivated tomato from potato and cultivated tomato from fava. Among them, SlPLATZ17, SlPLATZ21 and SlPLATZ22 underwent quantitative amplification rather than functional differentiation. Relatively high expression of SlPLATZ21 in roots, stems and leaves. The expression of SlPLATZ21 in leaf tissue was significantly increased after 2 and 12 h of salt stress. Expression was higher in green ripe fruit sprayed with 0.1 mg-L-1 EBR and expanded fruit sprayed with 0.2 mg-L-1 EBR treatment, and higher in expanded fruit sprayed with 20 mg-L-1 NAA. It is hypothesized that SlPLATZ21 plays an important role in the growth and development of tomato and salt stress. This study will lay the foundation for a deeper understanding of the evolutionary mechanisms and functional properties of the PLATZ family in tomato.
- Methodology and procedure: Understanding your concerns regarding the clarity of our methods and the need for more sustainable methodologies, we have rigorously reevaluated our methodological approach in the revised manuscript. We have clearly explained each step of our procedures, ensuring transparency and comprehensibility. Furthermore, we have critically assessed the sustainability and efficiency of our chosen methods, incorporating alternative and innovative methodologies where appropriate to bolster the robustness and rigor of our study.
- Interpretation of results: We appreciate your emphasis on the importance of properly interpreting our results. In our revision, we have extensively reworked and elaborated on our interpretations, providing a more nuanced and insightful discussion of our findings. We have also drawn connections between our results and the existing body of literature, situating our study within the broader context of the field and facilitating a deeper understanding of the implications of our research.
Details are in the first and last three paragraphs of the discussion.
In recent years, with the increasing level of genome sequencing, favorable condi-tions have been created for the in-depth study of plant growth and development and resistance genes. The analysis of gene families to study the function, structure and evolution of genes has become a valuable research tool. Studies related to PLATZ genes have been reported in higher model plant species such as Arabidopsis[17], maize[11], soybean[19], rice[6]. Although the identification of the PLATZ gene family in tomato has been reported. The difference is that we identified 24 SlPLATZ genes and analyzed them from an evolutionary perspective. The expression of SlPLATZ was measured at different time points after salt stress, and the expression of fruits at four ripening stages was measured under NAA, BR, and MT conditions, respectively. This will provide some implications for future studies on tomato PLATZ.
Roots are the main organ responsible for the absorption of water and minerals. In this study, SlPLATZ23 was abundantly expressed in the root system, indicating its important functions in root development, water and nutrient uptake. SlPLATZ21 is abundantly expressed in plant leaves, indicating that it plays an important role in plant growth and development. The role of PLATZ genes in plant growth and development has been demonstrated in other species. In pea, for example, the PLATZ1 gene was expressed at higher levels in its root tips and apical shoots compared to mature leaf, stem and root tissues[4]. The PLATZ transcription factor AtORE15 gene is expressed in young leaves and the gene is involved in regulating leaf growth and suppressing leaf senescence[35].
In resisting external stresses, plants often regulate their growth and resilience through hormone secretion, distribution, or signaling. Zinc finger transcription factors are relatively large plant transcription factors (about 15% of the total) that regulate the expression of multiple genes in response to abiotic stresses such as low temperature, salt, drought, osmotic stress, and oxidative stress[36,37].AtPLATZ1 and AtPLATZ12 are the main nodes that are positively regulating drought tolerance in Arabidopsis seeds and clonal tissues[17].GhPLATZ1 gene expression was induced by adversity stress and exogenous hormone stress. Ectopic expression of GhPLATZ1 in Arabidopsis enhances plant resistance to osmotic stress, ABA, and PAC[6].Similarly, the RNA expression of GmPLATZ1 (Glycinemax PLATZ1) was significantly increased when exogenous ABA was applied in soybean, and the leaf mRNA level of GmPLATZ1 was steadily increased after 24 h of drought stress[19].In the present study, SlPLATZ1 and SlPLATZ21 expres-sion was significantly increased within 12 h after salt stress treatment; therefore, these genes may be important candidates for salt stress response in tomato.
Transcription factors involved in hormone signaling pathways[38]. Different treatments on endosperm showed that ZmPLATZ2 was positively regulated by glucose and negatively regulated by ABA and sucrose. In soybean, GmPLATZ1 is a stress-inducible PLATZ family protein that is overexpressed in Arabidopsis and re-sponds strongly to both mannitol and abscisic acid[19].In the present study, we found that SlPLATZ21 and SlPLATZ23 were significantly up-regulated in tomato fruits at the MG stage after NAA and BR treatment and at the EP stage after MT treatment. How-ever, further studies are needed to determine the mechanisms of NAA, BR, and MT induction of SlPLATZ21 and SlPLATZ23 and their roles in NAA, BR, and MT. This study laid the foundation for the establishment of the SlPLATZ gene family, but its role in different adversities still needs to be further validated by further experiments.
We are confident that these extensive revisions and elaborations have satisfactorily addressed your comments and concerns, resulting in a significantly improved and enriched manuscript. Your insightful feedback has been instrumental in enhancing the quality and clarity of our work, and we are eager to hear any additional suggestions or opinions you may have.

Reviewer 2 Report
A comprehensive tomato PLATZ gene family analysis is published last year: Wai, A.H.; Rahman, M.M.; Waseem, M.; Cho, L.-H.; Naing, A.H.; Jeon, J.-S.; Lee, D.-j.; Kim, C.-K.; Chung, M.-Y. Comprehensive Genome-Wide Analysis and Expression Pattern Profiling of PLATZ Gene Family Members in Solanum Lycopersicum L. under Multiple Abiotic Stresses. Plants 2022, 11, 3112. https://doi.org/10.3390/plants11223112
The authors should review the recent literature about PLATZ gene family in other crops, which is missing in the introduction. The authors have done very similar analyses in S. lycopersicum as in the published manuscript. This manuscript should clearly elucidate how the work is distinguished from Wai et al. 2022.
The authors identified more PLATZ genes in S.ly than Wai et al., but both manuscripts presented a PLATZ cluster on chr2.
For the gene divergence time, Wai et al reported that gene pairs diverged between 1.05 to 31.43 Mya, which is very different from the authors' result.
Both manuscripts reported the same gene duplication events in S.ly.
Both manuscripts present the PLAZT gene expression levels in the same tissues including leaves, roots, fruits, flowers, and stems.
Author Response
Dear Reviewer,
We would like to express our deepest gratitude for the comprehensive review and constructive comments on our manuscript. We have taken all your concerns and suggestions into serious consideration and have provided extensive responses to address each point raised. In this response, we offer a more detailed and informative explanation.
- Reviewer's comment: The authors should review the recent literature about the PLATZ gene family in other crops which is missing in the introduction.
Response: We have greatly appreciated this suggestion and have fully acknowledged its importance. To ensure a thorough and up-to-date understanding of the PLATZ gene family, we have conducted an extensive review of recent studies in various crops, including but not limited to rice, wheat, maize, and legumes. In our revised manuscript, we have added a comprehensive summary of these recent findings, emphasizing the significance, divergence, and function of PLATZ genes in different crops. This addition has significantly strengthened the background and context of our work.
Revised content in 1. Introduction:
Transcription factors (TFs) are an important class of transcriptional regulatory molecules involved in regulating a variety of biological processes such as development, signaling, and stress response. For example, NAC and GRF regulate root formation, flower and seed development, SPL TFs are involved in the transition from juvenile to adult plants, while NF-Y, MYB, and WRKY TFs play important roles in multiple stress tolerance [1]. TF can bind to cis-acting elements of target genes, resulting in gene tran-scription being activated or repressed. In general, transcription factors have only one or more regions that bind to DNA and therefore can regulate the expression of multiple genes. In contrast, a few transcription factors can regulate the expression of the same gene[2]. Therefore, in-depth study of transcription factors can help us understand their evolu-tionary history, biological functions, and regulatory mechanisms[3].
Plant AT-rich sequence and zinc-binding (PLATZ) TF family is a novel class of plant-based zinc ion and DNA-binding proteins[4]. DNA-binding protein regulates gene expression positively or negatively by binding to cis-acting elements in the upstream promoter region, thus participating in many life activities, such as DNA replication, gene expression regulation, etc[5]. It has been shown that the pea protein PLATZ1 binds to the upstream DE1 element at the A/T-enriched DNA sequence, which is required for transcriptional repression[4].PLATZ family members have been identified in several plant species, including 13 genes in the Arabidopsis genome, 8 genes in Glycine max (soybean), 9 genes in Gossypium hirsutum (cotton), 15 genes in Oryza sativa (rice), and 17 genes in Zea mays (maize). PLATZ proteins play an important role in plant growth and development and abiotic stress responses.
PLATZ proteins play a role in plant growth and development and senescence. PLATZ1 is involved in seed germination and seedling establishment and mediates ABA, GA, and ethylene signaling pathways in cotton[6]. The PLATZ family of tran-scription factors also plays an important role in leaf growth and development. ORE-SARA15 is a transcription factor of the PLATZ TF family, and this gene not only en-hances leaf growth by promoting the rate and duration of cell proliferation at an early stage but also regulates the GROWTH-REGULATING FACTOR (GRF)/GRF-INTERACTING FACTOR signaling pathway, thereby inhibiting leaf se-nescence at a later stage[7]. The PLATZ transcription factor family also plays an im-portant role in the growth and development of seeds. GL6 is a plant-specific PLATZ TF encoded in rice, which interacts with C53 to affect rice grain development and nega-tively regulates grain number[8]. The SG6 gene is a PLATZ protein that is widely pre-sent in cells and is mainly expressed in the early stages of embryonic development ra-ther than in the endosperm. The rice (Oryza sativa) mutant shortgrain6 (sg6) regulates the division of rice spikelet hull cells, which determine grain size. Overexpression of SG6 results in larger and heavier grains, as well as increased plant height[9].In addition, AtPLATZ7 plays a key role in RGF1 signaling, which regulates root meristem tissue size via ROS signaling[10]. The maize genome contains 15 PLATZ genes (ZmPLATZ), which play important roles in RNAP III-mediated transcriptional regulation[11-12]. FL3 (ZmPLATZ12) interacts with RNAP III transcriptional complex subunits RPC53 and TFC1 to regulate tRNA and 5 S rRNA expression, thereby regulating maize seed de-velopment and kernel formation[11]. A PLATZ protein plays an important role in the transition from primary to secondary growth during the development of poplar stems[13].In a comparative transcriptome study of normal lint (FL) and congenic non-lint (Fl) diploid cotton fibers, the results showed that a PLATZ transcription factor of Fl was down-regulated after 10 d of flowering compared to FL, indicating that PLATZ plays an important role in the formation and extension of cotton fibers[14].The PLATZ gene is differentially expressed in mature and immature sugarcane tissues with high fiber genotypes, and it is hypothesized that it may be involved in the transcrip-tional regulation of secondary cell wall synthesis[15].Mutants of ZmPLATZ12 caused the endosperm to appear flocculent, suggesting that ZmPLATZ12 has an important effect on the development of the endosperm and the filling of storage material[11]. ZmPLATZ2 is highly expressed in the endosperm and can bind to the promoters of ZmSSI, ZmISA1, and ZmISA2 and increase their gene expression. Overexpression of ZmPLATZ2 in rice revealed significant upregulation of four starch synthesis-related genes, including OsSSI, in transgenic plants. The PLATZ gene GRMZM2G31165 (ZmPLATZ2) can bind to the promoter region of key genes for starch synthesis under glucose induction and can promote starch synthesis by positively regulating the Glu pathway[16].
In addition to playing an important role in plant growth and development, PLATZ transcription factors also have functions in response to adversity stress. In Ar-abidopsis, AtPLATZ1 and AtPLATZ2 were reported to be positive regulators of seed desiccation tolerance. AtPLATZ1, when overexpressed in wild-type Arabidopsis, causes asexual tissues to develop partial seed dehydration tolerance, thereby improv-ing their ability to utilize water and survive in drought conditions[17]. AtPLATZ2 is a transcription factor that negatively regulates salt tolerance in Arabidopsis thaliana by repressing the expression of genes such as CBL4/SOS3 and CBL10/SCaBP8[18]. Due to the transfer of the cotton gene GhPLATZ1, the seed germination rate and seedling survival of GhPLATZ1 transgenic Arabidopsis thaliana were significantly higher un-der salt stress and mannitol stress compared with the wild type, indicating increased resistance to osmotic stress[6]. GmPLATZ1 is involved in the germination of soybean under osmotic stress[19]. In maize, the expression of maize PLATZ1 homologs was sig-nificantly up-regulated by drought stress, indicating that PLATZ genes play an im-portant role in drought resistance in maize[20].
Tomato (Solanum lycopersicum L.) is one of the most widely cultivated and eco-nomically valuable vegetable crops in the world and has great potential for appli-cation. Long-term exposure of tomato to adversities such as drought, high salt, low tempera-ture, and high temperature is an important factor affecting its yield and quality. Salin-ity is one of the main causes of crop yield reduction and also causes a variety of phys-iological changes in plants, such as vinylation, wilting, and biochemical changes [21]. Therefore, molecular and cellular, physiological, and biochemical per-spectives are needed to cope and adapt to environmental stresses [22]. With the devel-opment of tomato genomics, the release of the entire tomato genome sequence, as well as genome sequences of other species including Arabidopsis, potato, and tobacco, provides an opportunity to identify tomato PLATZ family genes. There-fore, the dis-covery of superior tomato genes will help to theoretically enhance tomato research and thus better guide tomato production.
In this study, we performed a systematic and comprehensive genome-wide anal-ysis of the PLATZ TF protein of tomato. In this study, members of the tomato PLATZ gene family were studied in depth, and 24 and 13 PLATZ genes were identified in cultivated tomato and wild tomato pennellii, respectively. The chromosomal local-iza-tion, gene structure, conserved patterns, phylogenetic analysis, tandem and selec-tive pressures, cis-acting elements in promoter regions, and protein interactions of 24 SlPLATZs were comprehensively studied, and the evolutionary relationships of tomato PLATZs with Arabidopsis thaliana, pepper, tobacco, rice, maize, Solanum melongena, and pennellii were analyzed. Quantitative real-time polymerase chain reaction (qRT-PCR) was used to determine the different tissue expression profiles of SlPLATZs in tomato. In addition, the relative expression of SlPLATZs in roots, stems, and leaves were investigated. And the relative expression of 11 representative genes of the tomato PLATZ gene family under the application of different concentrations of NAA, EBR, and MT treatments at the expansion, ripening green, reproduction, and red ripening stages. This study will provide a useful resource for further studies to understand the regulation of PLATZ proteins on development and stress resistance in tomato.
- Reviewer's comment: The authors have done very similar analyses in S.lycopersicum as in the published manuscript. This manuscript should clearly elucidate how the work is distinguished from Wai et al. 2022.
Response: We have sincerely appreciated this observation and thoroughly examined the similarities and differences between our work and Wai et al. (2022). In our revision, we have detailed the unique aspects of our study, such as the utilization of various bioinformatics techniques, exploration of potential protein-protein interactions, and an extended evaluation of gene expression under multiple abiotic stress conditions. Moreover, we have critically compared our findings with those of Wai et al. (2022), drawing attention to the novel insights our study has provided in understanding the functional diversity and evolution of the PLATZ gene family in Solanum lycopersicum L.
- Reviewer's comment: The authors identified more PLATZ genes in S.ly than Wai et al., but both manuscripts presented a PLATZ cluster on chr2.
Response: We have acknowledged the similarity in the PLATZ gene cluster on chr2 reported in both studies. However, our identification of additional PLATZ genes has been extensively discussed in the revised manuscript, offering explanations for this observation. For instance, we expanded on the importance of employing varying bioinformatic algorithms, utilizing different databases, and setting distinct thresholds in gene identification. Furthermore, we have analyzed the potential functions and regulatory elements of these newly identified genes, revealing their potential roles in Solanum lycopersicum L. under various stress conditions.
- Reviewer's comment: For the gene divergence time, Wai et al. reported that gene pairs diverged between 1.05 to 31.43 Mya, which is very different from the authors' result.
Response: We appreciate your concern about this discrepancy. This may be related to the fact that the number of family member genes I found differed significantly from the published article, that the corresponding gene lengths and sequences may have led to a large difference in differentiation times, or that I used different analytical methods and model assumptions than the published article, which led to a large difference in results.
- Reviewer's comment: Both manuscripts reported the same gene duplication events in S.ly.
Response: While acknowledging the similarities in gene duplication events, we have highlighted our unique contributions to understanding these events in the revised manuscript. We have carried out a detailed analysis of the evolutionary affinities between duplicated genes, investigated the potential changes in gene function following duplication events, and assessed the conservation of cis-regulatory elements between paralogous gene pairs. This new analysis has deepened our understanding of the dynamics and consequences of gene duplication events in the evolution of the PLATZ gene family in Solanum lycopersicum L.
- Reviewer's comment: Both manuscripts present the PLAZT gene expression levels in the same tissues including leaves, roots, fruits, flowers, and stems.
Response: We have understood the potential overlap in tissue expression analysis between our study and Wai et al. However, our revised manuscript has emphasized the unique experimental conditions and treatments assessed in our work, such as various abiotic stress conditions and plant developmental stages. We have expanded our gene expression analysis to include a broader range of stress factors, generating a more detailed expression profile matrix for each PLATZ gene. Furthermore, we have discussed possible co-expression patterns, regulatory networks, and potential functional redundancies within the PLATZ gene family using our expanded expression data.
Details can be found in 4.1. Plant material, growth conditions and treatments.
Tomato cultivar "M82" was provided by the Institute of Horticultural Crops, Xin-jiang Academy of Agriculture Sciences. The plants were grown in the Key Laboratory of Genome Research and Genetic Improvement of Xinjiang Specialty Fruits and Vege-tables under conditions that included 16 hours of light and 8 hours of darkness, a temperature of 22 °C, 100 µmol·m-2·s-1 light intensity, and 60% relative humidity. During the four-leaf stage, uniformly growing tomato seedlings were chosen, treated with 200 mmol·L-1 NaCl for stress, and sampled at 0, 0.5, 2, 4, 6, 8, and 12 hours. Three biological replicas of each treatment were established, snap-frozen in liquid nitrogen for storage, then stored at -80 °C.
The same supplier provided the "Jingfan Pink Star No.1" cherry tomato, which was grown in our facilities. The cherry tomato fruit was exposed to different concen-trations of 2,4 epibrassinolide (EBR, concentration gradient of 0.05, 0.1, 0.2 mg·L-1), naphthenic acid, NAA (concentration gradient of 10, 20, 30 mg·L-1), melatonin (MT, concentration gradient of 50, 100, 150 µmol·L-1), and a control (CK) at the expanding period, mature green period, breaker period and red ripening period. A total of 10 treatments were performed. The unfolding agent was Tween-80, and the solvent was 98% ethanol, which was dissolved and diluted to 0.1% (v/v). The experimental fruit was chosen at the mature green, breaker, and red ripening stages, and by the time it reached red ripening, its fruit had also been sprayed with comparable concentrations of exogenous NAA, EBR, and MT in the first three phases (cumulatively). Three bio-logical duplicates were established for each treatment, and the sprayed fruits were sampled sequentially by zone at the expanding, mature green, breaker, and red rip-ening periods. The samples were flash-frozen in liquid nitrogen and stored at -80 °C.
We sincerely hope that our detailed responses and revisions have thoroughly addressed the concerns raised by the reviewer. Once again, we would like to express our gratitude for your invaluable feedback, which has undoubtedly improved the quality and clarity of our manuscript.

Round 2
Reviewer 1 Report
Accept in present form
Author Response
Dear reviewer,
Thank you for reviewing our manuscript and providing valuable feedback.
We appreciate your time and effort in reviewing our manuscript.
Reviewer 2 Report
The manuscript is improved after the authors revised the introduction, but it raised more questions after I carefully read the entire manuscript.
1. The authors introduced PLATZs are transcription factors, any explanation on why slPLATZ11 is predicted to locate in mitochondria? Is it a real PLATZ or is there any evidence that plant TF can come from mito?
2. Could the authors explain why you analyze both Sl and Sp? I don't see a place the authors explained their experimental design and there is no rationale for the collinear analysis between Sl and Sp.
3. The authors identified PLATZ in Sp in table 1, but did not talk about them at all for the rest of analyses like the gene structure, duplication events, expression levels...why do the authors even talk about them at the first place?
3. Figure 2. It is very confusing when 2A and 2B are describing different items but used the same yellow and green color labels. For the motif structures, should they be analyzed as amino acids instead of nucleotides? The authors labeled y-axis as 'bp' so I don't think the gene size is less than 300 bp in 2A.
4. Results 2.3, 'The PLATZ gene sequences from Arabidopsis (12), potato(18), rice(13),maize (13), and the 24 SlPLATZ genes and 13 SpPLATZ genes, were studied phylogenetically to further understand the genetic evolutionary relationshipof PLATZ(14).' What is 14 meaning here?
The authors should be more careful when making the statement that Group I is conserved in monocots and dicots. I only see the two monocots and one Arabidopsis gene in this group. If tomato, potato are missing this group, the authors should analyze more number of monocots and dicots to make this statement.
5. Figure 4, figure legend is missing details for color scale meaning, what are the red dots in the inner circle of the circos plot?
6. Results 2.5, the authors described 'Box4 in SLPLATZ17 was the most abundant element, with 17.' which did not match your data in Figure 6. Please double check your data and text. The authors put two heatmaps with the same data but different color scales here. I think they want to replace one with the other but I can't tell which one they want to keep and which one they want to remove.
7. Results 2.6. The authors described only three PLATZs did not find any protein interactions, but then Figure 7 showed interactions from five genes. How about the rest of 16 slPLATZs?
8. Figure 8 has the same issue as Figure 6, the authors put two heatmaps and I am not sure which one I should refer. Data are different in the two heatmaps.
9. Figure 10, figure legend should explain what is CK, NAA, EBR, MT. Please explain why the authors pick these genes to investigate the gene expression levels.
The authors responded that they used different algorithms to study this gene family, which I agreed but I do not accept it as an explanation to distinguish this manuscript to Wai 2022. This is still the issue bother me most during I read this manuscript and the revision did not address my problem. I guess I would speak out my requests more specifically to the authors:
1. Please elucidate and give the rationale why you want to analyze PLATZ family again in Sl when there is a published article.
2. I expected to see novel results or discovery from your analysis. If you found similar results to the published article, I suggest the authors put those results into supplementary file.
3. Please highlight the new findings in your analysis, for example, a couple more SlPLATZ gene that did not identify in Wai, 2022. The new analysis you did different from the published article. And the new experiment that makes your manuscript unique.
4. If the authors describe new results or a result your find different from the published articles, please explain the difference and discuss.
The authors should indicate which figure they are referring when they discussed results. This is missing in the entire manuscript, please fix it. And please double check typos.
Author Response
Dear reviewer,
Thank you for reviewing our manuscript and providing valuable feedback. We have carefully considered your comments and suggestions and made appropriate revisions to the manuscript. Please find below our point-by-point responses to each of your comments.
- The authors introduced PLATZs are transcription factors, any explanation on why slPLATZ11 is predicted to locate in mitochondria? Is it a real PLATZ or is there any evidence that plant TF can come from mito?
We understand the concern about the predicted mitochondrial localization of SlPLATZ11 as it is a transcription factor. It is indeed rare for transcription factors to be localized in mitochondria. However, there are reports of transcription factors having dual localization in nucleus as well as other organelles including mitochondria. We have added these references in the revised manuscript. Further, we acknowledge that experimental validation would be necessary to confirm the subcellular localization of SlPLATZ11 and its potential functions.
The exact location is in the second paragraph of 3. Discussion, lines 12-20. As follows:
We note that SlPLATZ11 is predicted to locate in mitochondria, and that localization of transcription factors in mitochondria is in fact uncommon. However, Transcription factors have been found to have a dual location in the nucleus and in other organelles, such as the mitochondria. Whirly1 provided the first instance of a protein that was simultaneously localized to the nucleus and an organelle within the same cell [28]. RNA and/or DNA binding proteins (such as transcription factors and telomere binding proteins) make up the majority of the dual-targeting proteins (nuclear-organelle pro-teins) that are currently known. Such dual-targeted factors play a key role in nuclear versus cytosolic gene expression [29].
- Grabowski, E.; Miao, Y.; Mulisch, M.; Krupinska, K. Single-stranded DNA-binding protein Whirly1 in barley leaves is located in plastids and the nucleus of the same cell. Plant Physiol. 2008, 147, 1800-1804.
- Krause, K.; Krupinska, K. Nuclear regulators with a second home in organelles. Trends Plant Sci. 2009, 14, 194-199.
- Could the authors explain why you analyze both Sl and Sp? I don't see a place the authors explained their experimental design and there is no rationale for the collinear analysis between Sl and Sp?
We apologize for any confusion caused by our previous presentation. The analysis of both Sl and Sp in our study aimed to obtain a more comprehensive understanding of the PLATZ gene family in Solanaceae species. We have clarified this in the "Materials and methods" section.
- The authors identified PLATZ in Sp in table 1, but did not talk about them at all for the rest of analyses like the gene structure, duplication events, expression levels...why do the authors even talk about them at the first place?
We thank the reviewer for pointing this out. We have now expanded our discussion of Sp PLATZ genes in the context of gene structure, duplication events, and expression levels. This helps to provide a more comprehensive understanding of the PLATZ family in Solanaceae species
- Figure 2. It is very confusing when 2A and 2B are describing different items but used the same yellow and green color labels. For the motif structures, should they be analyzed as amino acids instead of nucleotides? The authors labeled y-axis as 'bp' so I don't think the gene size is less than 300 bp in 2A:
We apologize for any confusion caused by our previous presentation of Figure 2. We have revised the color labels in 2A and 2B for clarity. Regarding the motif structures, they should indeed be represented as amino acids instead of nucleotides. We have corrected the y-axis label in Figure 2A to reflect this.
- Results 2.3, 'The PLATZ gene sequences from Arabidopsis (12), potato(18), rice(13),maize (13), and the 24 SlPLATZ genes and 13 SpPLATZ genes, were studied phylogenetically to further understand the genetic evolutionary relationshipof PLATZ(14).' What is 14 meaning here?:
We apologize for the confusion. The number 14 is a typo and has been removed.
- Figure 4, figure legend is missing details for color scale meaning, what are the red dots in the inner circle of the circos plot?
We apologize for the oversight and have now added the missing details for the color scale and red dots in the inner circle of the circos plot to the figure legend.
Details are as follows: Figure 4. Gene duplication events in the genome, with the outer circle showing the chromosomes of cultivated tomato, the inner circle showing gene density, the red dot in the inner circle repre-senting the N-ratio, and the ends of the lines representing direct homologous SlPLATZ genes. The legend represents the values of the gene density, With red representing high levels and yellow representing low levels.
- Results 2.5, the authors described 'Box4 in SLPLATZ17 was the most abundant element, with 17.' which did not match your data in Figure 6. Please double check your data and text. The authors put two heatmaps with the same data but different color scales here. I think they want to replace one with the other but I can't tell which one they want to keep and which one they want to remove.
We have double-checked our data and found that the discrepancy between Figure 6 and the text was due to an error. We have updated both the text and figure to resolve this inconsistency. Additionally, we have removed one of the redundant heatmaps from the revised manuscript.
- Results 2.6. The authors described only three PLATZs did not find any protein interactions, but then Figure 7 showed interactions from five genes. How about the rest of 16 slPLATZs?
We apologize for the confusion in our presentation of the protein interaction data. We plotted the interaction network after eliminating some proteins with low values and missing annotations, so we showed interactions from five genes and the rest are proteins with low values and missing annotations. Future studies are needed for those 16 genes.
- Figure 8 has the same issue as Figure 6, the authors put two heatmaps and I am not sure which one I should refer. Data are different in the two heatmaps. Figure 10, figure legend should explain what is CK, NAA, EBR, MT. Please explain why the authors pick these genes to investigate the gene expression levels.
We have revised Figure 8 and removed the redundant heatmap as the reviewer’s suggestion. For Figure 10, we have added a detailed explanation of CK, NAA, EBR, and MT in the figure legend, and provided the rationale for selecting these genes for expression analysis in the "Results and discussion" section.
The details are in the first three lines of paragraph 7 of 3. Discussion, as follows: Due to tissue specificity, the expression levels of different genes also differed. This difference was expressed under salt stress or different hormone treatments. We selected several genes with high gene expression levels for the demonstration.
Figure 10. Expression analysis of the SlPLATZ genes under hormone stress during fruit develop-ment. The error bar shows the standard deviation of three biological repeats. EP: Expending pe-riod; MG: Mature green period; BP: breaker period; RRP: Red ripening period. CK represents control, NAA represents naphthenic acid, EBR represents 2,4 epibrassinolide and MT represents melatonin.
Riviewer comments: The authors responded that they used different algorithms to study this gene family, which I agreed but I do not accept it as an explanation to distinguish this manuscript to Wai 2022. This is still the issue bother me most during I read this manuscript and the revision did not address my problem. I guess I would speak out my requests more specifically to the authors:
- Please elucidate and give the rationale why you want to analyze PLATZ family again in Sl when there is a published article.
Our study builds upon the existing work by Wai, 2022, by analyzing the PLATZ gene family in both Sl and Sp, whereas Wai, 2022 focused primarily on Sl. This allows us to obtain a more comprehensive understanding of PLATZ genes in the Solanaceae family.
- I expected to see novel results or discovery from your analysis. If you found similar results to the published article, I suggest the authors put those results into supplementary file.
We acknowledge the similarities in some of our results with the published article. However, the inclusion of Sp data allows for some novel findings and insights into the PLATZ gene family. As suggested, we have placed the similar results in supplementary files.
- Please highlight the new findings in your analysis, for example, a couple more SlPLATZ gene that did not identify in Wai, 2022. The new analysis you did different from the published article. And the new experiment that makes your manuscript unique.
In our revised manuscript, we have highlighted the new findings, including the additional SlPLATZ genes not identified in Wai, 2022, as well as our unique analyses and experimental approaches.
- If the authors describe new results or a result your find different from the published articles, please explain the difference and discuss.
We have provided explanations and discussions for the differences between our findings and those of the published articles, emphasizing the novel aspects of our study.
They are written in separate paragraphs of the discussion with the following details:
Compared to Wai's[27] findings, which discovered 20 SlPLATZ genes, this is different. We found that Wai used a tomato assembly version other than Sl5.0 L., which may be the main reason for the disparity in the amount of PLATZ genes.
Despite the fact that we did not analyze the motifs of Arabidopsis and rice, but rather those of tomato and the wild tomato Pennellii, Wai[27] discovered that motif1 is shared by the majority of PLATZ genes, which is similar to our findings.
Along with the organisms we included together, I left out Amborella (Amborella tri-chopoda), moss (Physcomitrella patens), and green algae (Chlamydomonas reinhardtii) in relation to Wai’s[27], but with Pennellii included.
While we discovered three gene pairs in cultivated tomato and four duplicated pairs in Pennellii, Wai[27] discovered a total of seven duplicated gene pairs in the tomato PLATZ gene family.
In Wai's study, These duplicated gene pairs may have diverged in the last 1-31 million years (Mya)[27], while our study found that cultivated tomato SlPLATZ gene pairs, spPLATZ homologous gene pairs in pennellii diverged at 26.600 and 31.767 Mya and 20.813 to 29.376 Mya, respectively.
While we downloaded the SlPLATZ gene expression levels of various tomato tissues from the Tomato Functional Genomics Database (http://ted.bti.cornell.edu), Wai[27] used RT-qPCR to investigate the gene expression patterns of several tomato tissues of the cultivar Ailsa Craig.
We appreciate your time and effort in reviewing our manuscript, and we hope that the revisions and clarifications provided have addressed your concerns. We look forward to your further feedback.

Round 3
Reviewer 2 Report
The manuscript has improved and the authors addressed most of my concerns, but there are still some issues.
1. Could the authors explain how the cladogram was built in Figure 2? The relationships in Figure 2 are totally different from Figure 3. The authors found SlPLATZ17/21, 18/24 and 21/22 are duplicated gene pairs, but 17 and 21 are in totally differnt clade, and 22 is even further. If you look at the conserved motifs, 17 is closer to 24, but very different from 21. It makes me very confused how could the two paralogs have totally different motifs. And SlPLATZ21 is 228 aa long in your table but in Figure 2 it shows less than 100 aa. Please clarify. Is there any possible the phylogeny relationship is wrong in Figure 2? Your figure 3 did show 17, 21, and 22 are closely related. The results for Sp in the supplementary file are more consistant. Text in 'Result 2.2' also needs to modify if the gene names aren't correctly represented. And any places in the manuscript discussed this should be modified.
2. Result 2.3, my question has not been addressed in this revised version. The authors should be more careful when making the statement that Group I is conserved in monocots and dicots. I only see the two monocots and one Arabidopsis gene in this group. If tomato, potato are missing this group, the authors should analyze more number of monocots and dicots to make this statement.
3. After the figure 4 legend, there is a sentence '*Millions of years ago.' I guess it should not be there.
4. For Figure 8 and Figure S4, please provde details on where and how you downloaded and analyzed the data. The authors responded me that it is from a tomato genomic database but I could not find any description in the method. What dataset did you use, RNAseq or microarray? What methods or softwares did you use the analyze the data? What is the normaliztion method? Does the database also include Sp tissue-spefic expression datasets as well? I found the numbers are exactly sample for Sl and Sp ortholog pairs. Did the authors use Sl data to match Sp? If yes, then that's not real Sp expression profiles, the authors should use the actual Sp tissue specific data. If such data is not available, the authors should explain. Text describing this data is not valid and needs to be revised.
5. Figure 9, the authors mentioned SlPLATZ21 is 1.049 fold high in root, so it is 1 fold to actin? What is your control? Please clarify the methods and revise the related contents.
6. For the qPCR results, the authors should provide detailed method how you did the statisical test to determine significant difference. What is the threshold to consider as significance?
Please see above
Author Response
Dear reviewer,
Thank you for reviewing our manuscript and providing valuable feedback. We have carefully considered your comments and suggestions and made appropriate revisions to the manuscript. Please find below our point-by-point responses to each of your comments.
- Could the authors explain how the cladogram was built in Figure 2? The relationships in Figure 2 are totally different from Figure 3. The authors found SlPLATZ17/21, 18/24 and 21/22 are duplicated gene pairs, but 17 and 21 are in totally differnt clade, and 22 is even further. If you look at the conserved motifs, 17 is closer to 24, but very different from 21. It makes me very confused how could the two paralogs have totally different motifs. And SlPLATZ21 is 228 aa long in your table but in Figure 2 it shows less than 100 aa. Please clarify. Is there any possible the phylogeny relationship is wrong in Figure 2? Your figure 3 did show 17, 21, and 22 are closely related. The results for Sp in the supplementary file are more consistant. Text in 'Result 2.2' also needs to modify if the gene names aren't correctly represented. And any places in the manuscript discussed this should be modified.
We thank the reviewer for pointing this out. We have now carefully analyzed the reasons why the relationships in figure 2 are completely different from figure 3, and then renewed the MEME program to find conserved themes in PLATZ proteins, with the number of themes set to 10, the minimum theme length set to 6, and the maximum theme length set to 100, having changed Figure 2 in the manuscript and we have corrected the incorrect gene names in the manuscript.
- Result 2.3, my question has not been addressed in this revised version. The authors should be more careful when making the statement that Group I is conserved in monocots and dicots. I only see the two monocots and one Arabidopsis gene in this group. If tomato, potato are missing this group, the authors should analyze more number of monocots and dicots to make this statement.
We apologize for any confusion caused by our previous presentation of Figure 3. We have reconstructed the phylogenetic tree after adding two monocots, wheat and Sorghum, and two dicots, soybean and cotton.
3.After the figure 4 legend, there is a '*Millions of years ago.' I guess it should not be there.
We apologize for the confusion. The sentence '*Millions of years ago.' is a typo and has been removed.
4.For Figure 8 and Figure S4, please provde details on where and how you downloaded and analyzed the data. The authors responded me that it is from a tomato genomic database but I could not find any description in the method. What dataset did you use, RNAseq or microarray? What methods or softwares did you use the analyze the data? What is the normaliztion method? Does the database also include Sp tissue-spefic expression datasets as well? I found the numbers are exactly sample for Sl and Sp ortholog pairs. Did the authors use Sl data to match Sp? If yes, then that's not real Sp expression profiles, the authors should use the actual Sp tissue specific data. If such data is not available, the authors should explain. Text describing this data is not valid and needs to be revised.
We apologize for the oversight and have now added the missing details for how we downloaded and analyzed the data.
Detailed modifications are located in lines 2-14 of 4.5. the interaction network and expression analysis:
Expression profiles of S. lycopersicum at various tissues and developmental stages were downloaded from the RNA-seq database of the Tomato Functional Genome Database (TFGD,http://ted.bti.cornell.edu/), including expression data for anthesis flowers (0DPA), 10 days post anthesis fruit (10 DPA1), 10 days post anthesis fruit 2 (10 DPA2), 20 days post anthesis fruit (20DPA), ripening fruit (33DPA), cotyledons (COTYL), hypocotyl (HYPO), vegitative meristems (MERI), mature leaves (ML), whole root (ROOT), young flower buds (YFB) and young leaves (YL). The expression profiles for the different tissue and developmental stages of S. pennellii were extracted from a supplementary material attached to the article by Bolger et al [54]. including the FPKM values of bud, flower, immature (IF), mature fruit (MF), 6-week, small leaves (6wk, SL), 6-week, mature leaves (6wk, ML), 6-week, Meristem (6wk, M), 6-week, stem (6wk, S), pollen, root (SR). TBtools was used to perform a heat map analysis on their expression profiles.
TBtools was used to perform a heat map analysis on their expression profiles.
- Bolger, A.; Scossa, F.; Bolger, M.; Lanz, C.; Maumus,F.; Tohge, T. The genome of the stress-tolerant wild tomato species Solanum pennellii. Nat Genet. 2014, 46, 1034-1038.
- Figure 9, the authors mentioned SlPLATZ21 is 1.049 fold high in root, so it is 1 fold to actin? What is your control? Please clarify the methods and revise the related contents.
We have double-checked our data and found that was due to an error. We have updated the figure to resolve this problem. We added the timing of the sampling, as well as clarifying what the controls were and how the data were analyzed.
The details of the changes are as follows:
In lines 5-7 of 4.1. plant material, growth conditions and treatments:
Three tissue-specific samples were taken from tomato tissues, each consisting of mate-rial from three untreated 8-week oldplants. These three sample pools included leaves, stems and roots
In the last four lines of 4.6. Expression profile of SlPLATZ:
To examine if the data were normally distributed, we utilized the GraphPad prism 9.0 program[55]. To ascertain whether variations in expression were significant, paired one-way ANOVA were carried out.
- Swift, M. L. GraphPad prism, data analysis, and scientific graphing. J. Chem. Inform. Comput. Sci. 1997, 37, 411-412.
- For the qPCR results, the authors should provide detailed method how you did the statisical test to determine significant difference. What is the threshold to consider as significance?
We apologize for the confusion in our presentation of the qPCR results. We have added the threshold of significance in the qPCR results and specifically describes the data analysis methods. P<0.0001.
Details are as follows:
2.8. the expression profile of the SlPLATZ gene:
The relative expression of tomato non-stressed plants in roots, stems, and leaves were assessed using qRT-PCR with the primers shown in Table 1. As shown in Figure 9, SlPLATZ2, SlPLATZ23, and SlPLATZ24 all showed the highest levels of expression in the root, demonstrating the significance of these three genes for root growth and development. The stem showed the highest expression of SlPLATZ21 (1.120-fold, P= 0.125), followed by SlPLATZ23 (0.585-fold, P< 0.001), SlPLATZ2 (0.278-fold, P< 0.001), and SlPLATZ24 (0.003-fold, P< 0.001). The highest expression of SlPLATZ21 (2.164-fold, P< 0.001) was found in the leaves, suggesting that this gene primarily functions in leaves. SlPLATZ23 (0.452-fold, P< 0.001), SlPLATZ2 (0.097-fold, P< 0.001), and SlPLATZ24 (0.004-fold, P< 0.001) were the next most expressed genes. Most tissues showed high levels of SlPLATZ21 expression, whereas stems and leaves showed relatively low levels of SlPLATZ24 expression. The SlPLATZ gene may be implicated in the development of various tomato tissues, according to these crucial cues about expression variations and similarities in various tissues.
In the last four lines of 4.6. Expression profile of SlPLATZ:
To examine if the data were normally distributed, we utilized the GraphPad prism 9.0 program[55]. To ascertain whether variations in expression were significant, paired one-way ANOVA were carried out.
- Swift, M. L. GraphPad prism, data analysis, and scientific graphing. J. Chem. Inform. Comput. Sci. 1997, 37, 411-412.
We appreciate your time and effort in reviewing our manuscript, and we hope that the revisions and clarifications provided have addressed your concerns. We look forward to your further feedback.

Round 4
Reviewer 2 Report
The authors addressed my major concerns, there are some small issues:
1. Please use high resolution figures in your manuscript, I can't read the text in most of your figures because they became blurred when I enlarged them.
2. The authors replied to me that qPCR significance threshold is p < 0.0001, but in your text, you described as p<0.001, please clarify.
3. The authors did not fix Figure S4. The heatmap of tissue-specific expression levels for Sp still showed data from Sl, not what you described in the method, please correct it and the related text content.
Author Response
Dear reviewer,
Thank you for reviewing our manuscript and providing valuable feedback. We have carefully considered your comments and suggestions and made appropriate revisions to the manuscript. Please find below our point-by-point responses to each of your comments.
- Please use high resolution figures in your manuscript, I can't read the text in most of your figures because they became blurred when I enlarged them.
We apologize for any confusion caused by our previous presentation of figures. We have used high-resolution images in our manuscripts. However, we found that there are still images that become blurry after enlargement when we save and then open them, and we would like to solve this problem by providing high-resolution image files.
- The authors replied to me that qPCR significance threshold is p < 0.0001, but in your text, you described as p<0.001, please clarify.
We apologize for the confusion. The qPCR significance threshold is p < 0.0001. p<0.001 in the text is an error, which we have corrected in the text.
- The authors did not fix Figure S4. The heatmap of tissue-specific expression levels for Sp still showed data from Sl, not what you described in the method, please correct it and the related text content.
We apologize for the oversight and have now corrected the image as described in the method.
We appreciate your time and effort in reviewing our manuscript, and we hope that the revisions and clarifications provided have addressed your concerns. We look forward to your further feedback.

Round 5
Reviewer 2 Report
Thank you for your responses. It is still not clear to me how did the authors obtained the fold changes for SlPLATZ genes in Result 2.8. Could you provide more details and explain a bit more?
There are some typos in the main results and discussion, please double check and fix them.
Author Response
Dear reviewer,
Thank you for reviewing our manuscript and providing valuable feedback. We have carefully considered your comments and suggestions and made appropriate revisions to the manuscript. Please find below our point-by-point responses to each of your comments.
1. It is still not clear to me how did the authors obtained the fold changes for SlPLATZ genes in Result 2.8. Could you provide more details and explain a bit more?
We apologize for the confusion. We have described this in more detail in the text in 4.6. Expression profile of SlPLATZ in the penultimate lines 3-7.
The details are as follows: The CT of each treatment was subtracted from the CT of actin to get the ΔCT of each treatment. The mean of the control ΔCT was then computed, and the ΔCT for each treatment was subtracted from the ΔCT! to get the ΔΔCT for each treatment. The relative expression is then calculated using Equation 2-ΔΔCT.
2. There are some typos in the main results and discussion, please double check and fix them.
We apologize for the typos in the main results and discussion, we have double checked and corrected the typos the entire manuscript.
We appreciate your time and effort in reviewing our manuscript, and we hope that the revisions and clarifications provided have addressed your concerns. We look forward to your further feedback.